# Improved Risk Bounds with Unbounded Losses for Transductive Learning

## Abstract

In the transductive learning setting, we are provided with a labeled training set and an unlabeled test set, with the objective of predicting the labels of the test points. This framework differs from the standard problem of fitting an unknown distribution with a training set drawn independently from this distribution. In this paper, we primarily improve the generalization bounds in transductive learning. Specifically, we develop two novel concentration inequalities for the suprema of empirical processes sampled without replacement for unbounded functions, marking the first discussion of the generalization performance of unbounded functions in the context of sampling without replacement. We further provide two valuable applications of our new inequalities: on one hand, we firstly derive fast excess risk bounds for empirical risk minimization in transductive learning under unbounded losses. On the other hand, we establish high-probability bounds on the generalization error for graph neural networks when using stochastic gradient descent which improve the current state-of-the-art results.

## 1 Introduction

In the field of machine learning research, the analysis of stochastic behavior based on empirical processes is an essential component of learning theory, particularly in understanding and enhancing algorithm performance. The supremum of empirical processes plays a crucial role in various application scenarios, such as empirical process theory, Rademacher complexity theory, Vapnik–Chervonenkis theory, etc. In recent years, concentration inequalities for traditional suprema of empirical processes are fully established fields and have been well studied in the literature such as [36, 4, 5, 1, 24, 39, 12, 29]. All these inequalities based on the assumption of independent and identically distributed random variables. However, in many practical contexts, the i.i.d. assumption does not hold, such as when training and testing data are drawn from different distributions or when there is temporal dependence among data points. Such scenarios are prevalent in fields like visual recognition and computational biology, necessitating alternatives to Talagrand's inequality.

Another significant context in learning theory is transductive learning which was firstly introduced by [40]. In transductive learning, the training samples are independently and without replacement drawn from a finite population, as opposed to the classic model of independent and with replacement sampling. In this setting, the learning algorithm not only acquires a labeled training set but also receives a set of unlabeled testing instances, with the goal of accurately predicting the labels of the test points. This configuration naturally arises in numerous applications such as text mining, computational biology, recommendation systems, visual recognition, and malware detection. In these cases, the number of unlabeled samples often far exceeds that of labeled samples, and the cost of labeling the unlabeled samples is high. Consequently, the development of transductive algorithms that leverage unlabeled data to enhance learning performance has increasingly attracted attention.

In theoretical analysis of transduction learning, we need to sample without replacement, which leads to big challenge and has not been fully understood yet. [13] firstly extended the global Rademacher complexities into transductive learning and established the inequalities without replacement. [38] derived two concentration inequalities using Hoeffding's reduction method and the entropy method. Nevertheless, both [13] and [38] considered only bounded function. In real scenarios, where the maximum value of the function may be large and even unbounded, but the frequency of very large

values tends to be small. To the best of our knowledge, the analysis in unbounded functions random variables in transductive learning has not been studied yet.

In this paper, we focus on sampling without replacement with unbounded functions. We introduce a novel concentration inequality for empirical process upper bounds under the scenario of sampling without replacement, particularly for the case of unbounded functions. This represents the first attempt to discuss generalization performance for unbounded functions under the condition of sampling without replacement.

In Section 2, we provide the definition of the transductive learning set-up, including the basic notations and the discussion of two related transductive learning settings introduced by [40]. We also introduce the notations of the unbounded random variables used in the following sections.

Our new concentration inequalities for the case of unbounded functions are provided in Section 3, which are, to the best of our knowledge, the first concentration inequalities for sampling without replacement for classes of unbounded functions. Furthermore, we discuss two significant applications of the new inequalities: firstly, we derive high-probability fast excess risk bounds for unbounded loss in transductive learning based on local uniform convergence in Subsection 4.1; secondly, in Subsection 4.2, we provide generalization error bounds for Graph Neural Networks (GNNs) with unbounded loss when utilizing Stochastic Gradient Descent (SGD) which is better than the state-of-the-art work [37] when $m = o(N^{2/5})$. All the proofs in this paper are given in Appendix.

Our contributions are summarized as follows:

- We derive two novel concentration inequalities for suprema of empirical processes when sampling without replacement for classes of sub-Gaussian and sub-exponential functions, which is the first in transductive learning.

- We provide fast excess risk bounds for transductive learning considering Bernstein condition with unbounded losses. To the best of our knowledge, existing results do not provide fast rates in GNNs.

- Applying our inequalities, we obtain the generalization gap of GNNs for node classification task for stochastic optimization algorithm. In more detail, we establish high probability bounds of generalization error and test error under sub-Gaussian and sub-exponential losses. Thanks for considering the variance information, our results are better than [37] in some scenarios.

## 2 PRELIMINARIES FOR TRANSDUCTIVE LEARNING

In transductive learning, the learner is provided with $m$ labeled training points and $u$ unlabeled test points. The objective of the learner is to obtain accurate predictions for the test points. Two different settings of transductive learning were given by [41]. One assumes that both the training and test sets are sampled i.i.d. from a same unknown distribution and the learner is provided with the labeled training and unlabeled test sets. Another assumes that the set $\boldsymbol{X}_N$ consisting of $N$ arbitrary input points without any other assumptions regrading its underlying source is given. Then we sample $m \leq N$ objects $\boldsymbol{X}_m \subseteq \boldsymbol{X}_N$ uniformly without replacement from $\boldsymbol{X}_N$ which makes the inputs in $\boldsymbol{X}_m$ dependent. Finally, for each input $\mathbf{x} \in \boldsymbol{X}_m$, the corresponding output $Y$ from some unknown distribution $P(Y|X)$. Thus we obtain all the labels for the set $\boldsymbol{X}_m$, we denote the training set as $S_m = (\boldsymbol{X}_m, \boldsymbol{Y}_m)$. The remaining unlabeled set $\boldsymbol{X}_u = \boldsymbol{X}_N \backslash \boldsymbol{X}_m$, $u = N - m$ is the test set.

In this paper we study the second setting, as pointed out by [41], any upper generalization bound in the second setting can easily yield a bound for the first setting by just taking expectation. Note that related work [10, 14, 38] considers a special case where the labels are obtained from some unknown but deterministic function $\phi : \mathcal{X} \mapsto \mathcal{Y}$ so that $P(\phi(\mathbf{x})|\mathbf{x}) = 1$. We follow their assumption in this paper. Then the learner is a function model $f(\mathbf{w})$ w.r.t. the parameters $\mathbf{w}$ from some fixed hypothesis parameter space $\mathcal{W}$ which may not necessarily containing $\phi$. The choice of the learner based on both the labeled training set $S_m$ and the unlabeled test set $\boldsymbol{X}_u$. For brevity, we denote $\ell(\mathbf{w}; \mathbf{x}) = c(f(\mathbf{w}, \mathbf{x}), \phi(\mathbf{x}))$ w.r.t. the parameters $\mathbf{w}$ and the random variable $\mathbf{x}$, where $c : \mathcal{Y}^2 \mapsto \mathbb{R}_+$ is the cost function to measure the error of predicted label and real label on a point $X$. Then we can define the training error and test error of the learner as follows: $\hat{R}_m(\mathbf{w}) = \frac{1}{m} \sum_{\mathbf{x} \in \boldsymbol{X}_m} \ell(\mathbf{w}; \mathbf{x})$,

$R_u(\mathbf{w}) = \frac{1}{u} \sum_{\mathbf{x} \in \boldsymbol{X}_u} \ell(\mathbf{w}; \mathbf{x})$, where hat emphasizes the fact that the training (empirical) error can be computed from the data.

For technical reasons that will become clear later, we define the overall error to the union of the training and test sets as $R_N(\mathbf{w}) = \frac{1}{N} \sum_{\mathbf{x} \in \boldsymbol{X}_N} \ell(\mathbf{w}; X)$. The main goal of the learner in transductive setting is to select a proper parameters to minimizing the test error $R_u(\mathbf{w})$, which we will denote by $\mathbf{w}_u^*$. Since the labels of the test set examples are unknown, we can't compute $R_u(\mathbf{w})$ and need to estimate it based on the training sample $\boldsymbol{X}_m$ (and potentially using information from the features $\boldsymbol{X}_u$). A common choice is to replace the test error minimization by empirical risk minimization $\hat{\mathbf{w}}_m = \arg \min_{\mathbf{w} \in \mathcal{W}} \hat{R}_m(\mathbf{w})$ and to use it as an approximation of $\mathbf{w}_u^*$. For $\mathbf{w} \in \mathcal{W}$ we define the excess risk:

$$\varepsilon_u(\mathbf{w}) = R_u(\mathbf{w}) - \inf_{\mathbf{w}' \in \mathcal{W}} R_u(\mathbf{w}') = R_u(\mathbf{w}) - R_u(\mathbf{w}_u^*).$$

In the following sections, we establish some fundamental notations. We use $\| \cdot \|_2$ to represent the Euclidean norm of a vector and $\| \cdot \|$ to denote the spectral norm of a matrix. Throughout this study, we let $\mathcal{B}(\mathbf{w}'; r) \triangleq \{\mathbf{w} : \|\mathbf{w} - \mathbf{w}'\|_2 \leq r\}$, representing a ball with center vector $\mathbf{w}'$ and radius $r$. The gradient of the function $\ell$ with respect to its first argument is denoted as $\nabla \ell$. Next, we define the Orlicz norm to describe unbounded random variables.

**Definition 1** ([43]). *For $\alpha > 0$, define the function $\psi_\alpha : \mathbb{R}_+ \to \mathbb{R}_+$ with the formula $\psi_\alpha(x) = \exp(x^\alpha) - 1$. For a random variable $X$, define the Orlicz norm*

$$\|X\|_{\psi_\alpha} = \inf\{\lambda > 0 : \mathbb{E}\psi_\alpha(|X|/\lambda) \leq 1\}.$$

*Furthermore, a random variable $X \in \mathbb{R}$ is sub-Gaussian if there exists $K > 0$, such that $\|X\|_{\psi_2} \leq K$. A random variable $X \in \mathbb{R}$ is sub-exponential if there exists $K > 0$, such that $\|X\|_{\psi_1} \leq K$. A random variable $X \in \mathbb{R}$ is sub-Weibull if for $\forall \lambda > 0$, there exists $K > 0$, such that $\|X\|_{\psi_\alpha} \leq K$.*

**Remark 1.** Orlicz norm is a classical norm. By choosing an appropriate $\alpha$, we can define the tail distribution of random variables to different degrees using the Orlicz norm. This paper mainly discusses sub-Gaussian and sub-exponential distributions for loss functions. We use concentration inequality of the sum for sub-Weibull distribution during some proofs in applications, therefore, we provide this unified definition of unbounded random variables based on the Orlicz norm here.

## 3 Concentration Inequalities with Unbounded Losses

To gain the generalization error bounds for transductive learning with unbounded losses, we develop the novel concentration inequalities for suprema of empirical processes when sampling without replacement for unbounded functions.

We firstly introduce some necessary notations and settings. Let $\mathcal{C} = \{c_1, \ldots, c_N\}$ be some finite set. For $m \leq N$, let $\{X_1, \ldots, X_m\}$ and $\{X_1', \ldots, X_m'\}$ be sequence of random variables sampled uniformly with and without replacement from $\mathcal{C}$. Let $\mathcal{F}$ be a (countable[1] class of functions $f : \mathcal{C} \to \mathbb{R}$, such that $\mathbb{E}[f(X_1)] = 0$ for all $f \in \mathcal{F}$. It follows that $\mathbb{E}[f(X_1')] = 0$ since $X_1$ and $X_1'$ are identically distributed. Define the variance $\sigma^2 = \sup_{f \in \mathcal{F}} \mathbb{V}[f(X_1)]$. Note that $\sigma^2 = \sup_{f \in \mathcal{F}} \mathbb{E}[f(X_1)^2] = \sup_{f \in \mathcal{F}} \mathbb{V}[f(X_1')]$. Finally define that the supremum of the empirical process for sampling with and without replacement

$$Q_m = \sup_{f \in \mathcal{F}} \sum_{i=1}^m f(X_i), \quad Q_m' = \sup_{f \in \mathcal{F}} \sum_{i=1}^m f(X_i').$$

Concentration inequalities for sampling with replacement $Q_m$ have undergone extensive investigation, including the exploration of Talagrand-type inequality [36] and its variations as presented by [5, 4]. In the case of unbounded functions, certain studies, such as [1, 12] have established tail bounds through truncation methods and Talagrand-type inequalities for suprema of bounded empirical processes. Nevertheless, as of the current date, no bounds for the suprema of empirical processes

---

[1]Note that all results can be translated to the uncountable classes, for instance, if the empirical process is separable, meaning that $\mathcal{F}$ contains a dense countable subset. Details can be referred in page 314 of [3] or page 72 of [5]

involving unbounded functions for sampling without replacement $Q'_m$ have been established in the literature.

Next, we will introduce the innovative concentration inequalities for the suprema of empirical processes under the condition of sampling without replacement. These new results will be established separately for sub-Gaussian and sub-exponential functions.

**Theorem 1. (Concentration inequality when sampling without replacement for classes of sub-Gaussian functions)** Assume that for all $c \in \mathcal{C}$, $\| \sup_{f \in \mathcal{F}} |f(c)| \|_{\psi_2} < \infty$, for any $\epsilon > 0$, we have the following inequality that

$$
\mathbb{P}\left\{Q'_m - (1+\eta)\mathbb{E}[Q_m] \geq \epsilon\right\} \leq 6 \exp\left(-\frac{\epsilon^2}{16(1+\beta)m\sigma^2 + 8C^2 \left\|\max_{1 \leq i \leq m} \sup_{f \in \mathcal{F}} f(X_i)\right\|_{\psi_2}^2}\right).
$$

We also have that for any $\delta \in (0, 1)$, with probability at least $1 - \delta$,

$$
Q'_m \leq (1+\eta)\mathbb{E}[Q_m] + \sqrt{\left(16(1+\beta)m\sigma^2 + 8C^2 \left\|\max_{1 \leq i \leq m} \sup_{f \in \mathcal{F}} f(X_i)\right\|_{\psi_2}^2\right) \log\frac{6}{\delta}},
$$

where $\eta, \beta$ are some positive constants and $C$ is a positive constants depending on $\eta, \beta$.

**Theorem 2. (Concentration inequality when sampling without replacement for classes of sub-exponential functions)** Assume for all $c \in \mathcal{C}$, $\| \sup_{f \in \mathcal{F}} |f(c)| \|_{\psi_1} < \infty$, for any $\epsilon > 0$, we have the following inequality that

$$
\mathbb{P}\left\{Q'_m - (1+\eta)\mathbb{E}[Q_m] \geq \epsilon\right\} \leq 2 \exp\left(-\frac{\epsilon^2}{16(1+\beta)m\sigma^2 + 48C^2 \left\|\max_{1 \leq i \leq m} \sup_{f \in \mathcal{F}} f(X_i)\right\|_{\psi_1}^2}\right).
$$

We also have that for any $\delta \in (0, 1)$, with probability at least $1 - \delta$,

$$
Q'_m \leq (1+\eta)\mathbb{E}[Q_m] + \sqrt{\left(16(1+\beta)m\sigma^2 + 48C^2 \left\|\max_{1 \leq i \leq m} \sup_{f \in \mathcal{F}} f(X_i)\right\|_{\psi_1}^2\right) \log\frac{2}{\delta}},
$$

where $\eta, \beta$ are some positive constants and $C$ is a positive constants depending on $\eta, \beta$.

**Remark 2.** Although the appearance of $\mathbb{E}[Q_m]$ may seem to be unexpected at first glance, it is usually desirable to control the concentration of a random variable around its expectation. Fortunately, it has been demonstrated in [38] that for $m = o(N^{2/5})$, the difference $\mathbb{E}[Q_m] - \mathbb{E}[Q'_m]$ is bounded by $\sqrt{m}$. Consequently, our theorems can be employed to effectively manage the deviations of $Q'_m$ from its expectation $\mathbb{E}[Q'_m]$ at a fast rate.

In fact, we draw inspiration from the proof presented in [38] and use Hoeffding's reduction method to build the connection between the sequences of random variables sampling with and without replacement. However, extending the results to the classes of sub-Gaussian and sub-exponential functions presents challenges. On one hand, the classical truncation technique yields tail bounds, nonetheless we need to combine the sequences of random variables sampling with and without replacement using moment generating functions while ensuring their convexity. This is crucial as Hoeffding's reduction method requires convexity. On the other hand, the introduction of the unbounded assumption introduces an additional term, which complicates the construction of convex moment generating functions (MGF) and the application of cheronff's method.

## 4 GENERALIZATION BOUNDS FOR TRANSDUCTIVE LEARNING

Our concentration inequalities have broad applications and can serve as an important tool in learning theory when considering sampling without replacement for classes of sub-exponential functions. In this section, we will provide two examples to illustrate the risk bounds in transductive learning.

## 4.1 Fast Excess Risk Bounds for Transductive Learning with Unbounded Losses

We apply our newly concentration inequalities to give fast excess risk bounds for transductive learning on ERM with unbounded losses, which is, to the best of our knowledge, the first results. We mainly follows the traditional technique called "local Rademacher complexity" developed by [2]. We introduced the definition of Rademacher complexity for completeness.

**Definition 2** (Rademacher complexity [44]). *For a function class $\mathcal{F}$ that consists of mappings from $\mathcal{Z}$ to $\mathbb{R}$, define*

$$\mathfrak{R}\mathcal{F} := \mathbb{E}_{\mathbf{x},v} \sup_{f \in \mathcal{F}} \frac{1}{n} \sum_{i=1}^{n} v_i f(\mathbf{x}_i) \quad and \quad \mathfrak{R}_n \mathcal{F} := \mathbb{E}_v \sup_{f \in \mathcal{F}} \frac{1}{n} \sum_{i=1}^{n} v_i f(\mathbf{x}_i),$$

*as the Rademacher complexity and the empirical Rademacher complexity of $\mathcal{F}$, respectively, where $\{v_i\}_{i=1}^{n}$ are i.i.d. Rademacher variables for which $\mathbb{P}(v_i = 1) = \mathbb{P}(v_i = -1) = \frac{1}{2}$.*

Since Rademacher complexity could be bounded by a computable covering number of $\mathcal{F}$ via Dudley's integral bound [35], we give the definition of covering number for completeness as well.

**Definition 3** (Covering number [44]). *Assume $(\mathcal{M}, \mathrm{metr}(\cdot, \cdot))$ is a metric space, and $\mathcal{F} \subseteq \mathcal{M}$. The $\varepsilon$-convering number of the set $\mathcal{F}$ with respect to a metric $\mathrm{metr}(\cdot, \cdot)$ is the size of its smallest $\varepsilon$-net cover:*

$$\mathcal{N}(\varepsilon, \mathcal{F}, \mathrm{metr}) = \min\{m : \exists f_1, \ldots, f_m \in \mathcal{F} \text{ such that } \mathcal{F} \subseteq \cup_{j=1}^{m} \mathcal{B}(f_j, \varepsilon)\},$$

*where $\mathcal{B}(f, \varepsilon) := \{\tilde{f} : \mathrm{metr}(\tilde{f}, f) \leq \varepsilon\}$.*

To calculate the covering number, we also need the following assumption.

**Assumption 1** (Entropy bounds). *The parameter class $\mathcal{W}$ is separable and there exist $\mathcal{C} \geq 1, K \geq 1$ such that $\forall \varepsilon \in (0, K]$, the $L_2(\mathbb{P})$-covering numbers and the universal metric entropies of $\mathcal{G}$ are bounded as $\log \mathcal{N}(\varepsilon, \mathcal{G}, L_2(\mathbb{P})) \leq \mathcal{C} \log (K/\varepsilon)$.*

**Remark 3.** Assumption 1 was widely adopted in fast learning rates in statistic learning [31, 30, 11]. In fact, if $\mathcal{W}$ has finite VC-dimension, then Assumption 1 is satisfied [3, 6]. Some literature such as [23] assume that the envelope function is sub-exponential, which is a much stronger assumption.

It will be convenient to introduce the following operators, mapping functions $f$ defined on $\mathbf{X}_N$ to $\mathbb{R}$:

$$Ef = \frac{1}{N} \sum_{i=1}^{N} f(\mathbf{x}_i), \mathbf{x}_i \in \mathbf{X}_N.$$

Assume that there is a function $\mathbf{w}_N^* \in \mathcal{W}$ satisfying $R_N(\mathbf{w}_N^*) = \inf_{\mathbf{w} \in \mathcal{W}} R_N(\mathbf{w})$. Define the excess loss class $\mathcal{F}^* = \{f : f(\mathbf{x}) = \ell(\mathbf{w}; \mathbf{x}) - \ell(\mathbf{w}_N^*; \mathbf{x}), \mathbf{w} \in \mathcal{W}\}$.

**Theorem 3.** *Assume that there is a constant $B > 0$ such that for every $f \in \mathcal{F}^*$ we have $Ef^2 \leq B \cdot Ef$. Suppose Assumptions 1 hold and the objective function $\ell(\cdot; \cdot)$ is sub-Gaussian. For any $\delta \in (0, 1)$, with probability $1 - \delta$,*

$$\varepsilon_u(\hat{\mathbf{w}}_m) = \mathcal{O}\left( \frac{N}{mu} \Big( \log m + \log u + \frac{N \log \frac{1}{\delta}}{m} + \frac{N \log \frac{1}{\delta}}{u} + \sqrt{\log N \log \frac{1}{\delta}} \Big) \right).$$

**Theorem 4.** *Assume that there is a constant $B > 0$ such that for every $f \in \mathcal{F}^*$ we have $Ef^2 \leq B \cdot Ef$. Suppose Assumptions 1 hold and the objective function $\ell(\cdot; \cdot)$ is sub-exponential. For any $\delta \in (0, 1)$, with probability $1 - \delta$,*

$$\varepsilon_u(\hat{\mathbf{w}}_m) = \mathcal{O}\left( \frac{N}{mu} \Big( \log m + \log u + \frac{N \log \frac{1}{\delta}}{m} + \frac{N \log \frac{1}{\delta}}{u} + \sqrt{\log^2 N \log \frac{1}{\delta}} \Big) \right).$$

**Remark 4.** By utilizing variance information and introducing the Bernstein condition, we present the first results for fast learning rates under unbounded losses. Applying our concentration inequalities under unbounded conditions to local Rademacher method is not a straightforward task. We need to skillfully separate variance term and the Orlicz norm term through inequalities while constructing a suitable partition. Similarly, when employing the localized approach, we need to create a slightly modified version for partition $E_m f$ which is affected by the Hoeffding's reduction method applied during the proof of our concentration inequalities given in Section 3.

## 4.2 IMPROVED BOUNDS OF GNNS WITH SGD

GNNs have achieved great success in practice, but research on the generalization performance of GNNs for node classification remains limited. In the real world, training nodes are sampled without replacement from the entire node set, and test nodes remain visible during training [13, 32], which perfectly fits the transductive learning setting.

The current state-of-the-art work on generalization error for graph node classification [37] was based on the concentration inequality for transductive learning provided by [13]. In this subsection, we aim to obtain a tighter generalization upper bound by applying our new concentration inequalities introduced in this paper.

Let's introduce some notations for GNNs firstly. Consider an undirected graph $\mathcal{G} = \{\mathcal{V}, \mathcal{E}\}$, where $\mathcal{V}$ represents a set of nodes and $\mathcal{E}$ represents the edges between these nodes. The graph has a total of $n = |\mathcal{V}|$ nodes. Each node corresponds to an instance denoted as $\mathbf{z}_i = (\mathbf{x}_i, y_i)$, comprising a feature vector $\mathbf{x}_i$ and a label $y_i$ from a space $\mathcal{Z} = \mathcal{X} \times \mathcal{Y}$.

Let $\mathbf{X}$ denote the feature matrix, where the $i$-th row $\mathbf{X}_{i*}$ represents the feature $\mathbf{x}_i$. The adjacency matrix is represented as $\mathbf{A}$, and the diagonal degree matrix is denoted as $\mathbf{D}$. Specifically, the diagonal entry $\mathbf{D}_{ii}$ is computed as the sum of the weights of the edges connected to node $i$. We introduce the normalized adjacency matrix $\tilde{\mathbf{A}} = (\mathbf{D}+\mathbf{I}_n)^{-\frac{1}{2}}(\mathbf{A}+\mathbf{I}_n)(\mathbf{D}+\mathbf{I}_n)^{-\frac{1}{2}}$, where $\mathbf{I}_n$ is the identity matrix of size $n \times n$, and $\sqrt{|\mathcal{Y}|}$ corresponds to the square root of the number of categories. This matrix accounts for self-loops and captures the graph's normalized connectivity structure, aiding in subsequent analyses. We limit the scope of the learner to a given GNN and let $\mathbf{w}$ be its learnable parameters. Given the isomorphism between $\mathbb{R}^{p \times q}$ and $\mathbb{R}^{pq}$, our analysis in this work focuses on the more concise vector space. To achieve this, we introduce a unified vector $\mathbf{w} = [\text{vec}\,[\mathbf{W}_1]; \ldots; \text{vec}\,[\mathbf{W}_H]]$ to represent the collection $\{\mathbf{W}_h\}_{h=1}^{H}$, where $\text{vec}[\cdot]$ denotes the vectorization operator that transforms a given matrix into a vector. In other words, $\text{vec}\,[\mathbf{W}] = [\mathbf{W}_{*1}; \ldots; \mathbf{W}_{*q}]$ for $\mathbf{W} \in \mathbb{R}^{p \times q}$. In this context, $\mathbf{W}_{*i}$ represents the $i$-th column of $\mathbf{W}$.

In this section, we apply the concentration inequalities presented in this paper to derive improved rates of the current optimal results [37] for GNNs with SGD (Algorithm 1). The initialization weight of the model is denoted as $\mathbf{w}^{(1)}$. We use $b_g$ to represent the supremum of the gradient when evaluated at the initialized parameters, defined as $b_g = \sup_{z \in \mathcal{Z}} \left\| \nabla \ell(\mathbf{w}^{(1)}; z) \right\|_2$. The activation function is represented by $\omega(\cdot)$.

We notice that since the full data $\boldsymbol{X}_N$ is given, then $R_N(\mathbf{w}) = \frac{1}{N} \sum_{i=1}^{N} \ell(\mathbf{w}; \mathbf{x}_i)$ is not a random variable. Also, for any training sample $\boldsymbol{X}_m$, the test error $R_u(\mathbf{w})$ can be expressed in terms of $R_N(\mathbf{w})$ and the training error $\hat{R}_m(\mathbf{w})$ as follows:

$$R_u(\mathbf{w}) = \frac{1}{u} \sum_{i=m+1}^{m+u} \ell(\mathbf{w}; \mathbf{x}_i) = \frac{1}{u} \left( (m+u)R_N(\mathbf{w}) - \sum_{i=1}^{m} \ell(\mathbf{w}; \mathbf{x}_i) \right) = \frac{m+u}{u} R_N(\mathbf{w}) - \frac{m}{u} \hat{R}_m(\mathbf{w}).$$

Thus, for any fixed $\mathbf{w} \in \mathcal{W}$, the quantity $R_u(\mathbf{w}) - \hat{R}_m(\mathbf{w}) = \frac{N}{u}(R_N(\mathbf{w}) - \hat{R}_m(\mathbf{w}))$, for any $\hat{\mathbf{w}}$, we have

$$R_u(\hat{\mathbf{w}}) - \hat{R}_m(\hat{\mathbf{w}}) \leq \sup_{\mathbf{w} \in \mathcal{W}} R_u(\mathbf{w}) - \hat{R}_m(\mathbf{w}) = \frac{N}{u} \sup_{\mathbf{w} \in \mathcal{W}} R_N(\mathbf{w}) - \hat{R}_m(\mathbf{w}).$$

Note that for any fixed $\mathbf{w} \in \mathcal{W}$, $\mathbb{E}_{\mathbf{x}}[R_N(\mathbf{w}) - \ell(\mathbf{w}; \mathbf{x})] = R_N(\mathbf{w}) - \mathbb{E}_{\mathbf{x}}\ell(\mathbf{w}; \mathbf{x}) = 0$, thus, we can use the transductive setting described in Section 3. Considering the function class $\mathcal{F}_{\mathbf{w}} := \{f_{\mathbf{w}} : f_{\mathbf{w}}(\mathbf{x}) = R_N(\mathbf{w}) - \ell(\mathbf{w}; \mathbf{x}), \mathbf{w} \in \mathcal{W}\}$ associated with $\mathcal{W}$. For fixed $\mathbf{w}$, $R_N(\mathbf{w})$ is not random, at the same time, centering random variable does not change its variance, so we have

$$\sigma_{\mathcal{W}}^2 = \sup_{f_{\mathbf{w}} \in \mathcal{F}_{\mathbf{w}}} \mathbb{V}[f_{\mathbf{w}}(\mathbf{x})] = \sup_{\mathbf{w} \in \mathcal{W}} \mathbb{V}[\ell(\mathbf{w}; \mathbf{x})] = \sup_{\mathbf{w} \in \mathcal{W}} \left( \frac{1}{N} \sum_{\mathbf{x} \in \boldsymbol{Z}_N} (\ell(\mathbf{w}; \mathbf{x}) - R_N(\mathbf{w}))^2 \right).$$

Using Theorem 1 and 2, we can obtain the results that hold without any other assumptions, expect for the classes of sub-Gaussian or sub-exponential functions on the learning problem

$$\sup_{\mathbf{w} \in \mathcal{W}} (R_N(\mathbf{w}) - \hat{R}_m(\mathbf{w})) \leq (1+\eta)E_m + 2\sqrt{\left( \frac{4(1+\beta)\sigma_{\mathcal{W}}^2}{m} + \frac{2C^2 \| \max_{\mathbf{x}} \sup_{f \in \mathcal{F}_{\mathbf{w}}} f_{\mathbf{w}}(\mathbf{x}) \|_{\psi_2}^2}{m^2} \right) \log \frac{6}{\delta}},$$

---

**Algorithm 1** SGD for Transductive Learning

---

**Input:** Initial parameter $\mathbf{w}^{(1)}$, step sizes $\{\eta_t\}$, training set $\{\mathbf{x}_i\}_{i=1}^{m+u} \cup \{y_i\}_{i=1}^m$.
**for** $t = 1$ **to** $T$ **do**
    Randomly draw $j_t$ from the uniform distribution over the set $\{j : j \in [m]\}$.
    Update parameters by
        $\mathbf{w}^{(t+1)} = \mathbf{w}^{(t)} - \eta_t \nabla\ell(\mathbf{w}^{(t)}; \mathbf{x}_{j_t})$.
**end for**

---

and

$$\sup_{\mathbf{w}\in\mathcal{W}} (R_N(\mathbf{w}) - \hat{R}_m(\mathbf{w})) \le (1+\eta)E_m + 4\sqrt{\left(\frac{(1+\beta)\sigma_{\mathcal{W}}^2}{m} + \frac{3C^2\|\max_{\mathbf{x}}\sup_{f\in\mathcal{F}_{\mathbf{w}}} f_{\mathbf{w}}(\mathbf{x})\|_{\psi_1}^2}{m^2}\right)\log\frac{6}{\delta}},$$

where let $\{\xi_1, \ldots, \xi_n\}$ be random variables sampled with replacement from $\boldsymbol{X}_N$ and denote

$$E_m = \mathbb{E}\left[\sup_{\mathbf{w}\in\mathcal{W}}\left(R_N(\mathbf{w}) - \frac{1}{m}\sum_{i=1}^m \ell(\mathbf{w}; \xi_i)\right)\right].$$

Next, we need to derive the upper bounds of $\sigma_{\mathcal{W}}^2$, $E_m$ and $\|\max_{\mathbf{x}}\sup_{f\in\mathcal{F}_{\mathbf{w}}} f_{\mathbf{w}}(\mathbf{x})\|_{\psi_\alpha}^2, \alpha = 1$ or $2$ in GNNs with SGD. We present the assumptions only used in this subsection.

**Assumption 2.** *Assume that there exists a constant $c_X > 0$ such that $\|\mathbf{x}\|_2 \le c_X$ holds for all $\mathbf{x} \in \mathcal{X}$ and there exists a constant $c_W > 0$ such that $\|\boldsymbol{W}_h\| \le c_W$, $h \in [H]$ for $\mathbf{w} \in \mathcal{W}$.*

**Remark 5.** Assumption 2 necessitates boundness of input features as discussed by [42] and the boundness of parameters during the training process, which is a common consideration in the generalization analysis of Graph Neural Networks (GNNs) [16, 28, 9, 15]. This assumption play a crucial role in the analysis of Lipschitz continuity and Hölder smoothness of the objective with respect to the parameters $\mathbf{w}$.

**Assumption 3.** *Assume that the activation function $\omega(\cdot)$ is $\tilde{\alpha}$-Höder smooth. To be specific, let $P > 0$ and $\tilde{\alpha} \in (0, 1]$, for all $\mathbf{u}, \mathbf{v} \in \mathbb{R}^d$,*

$$\|\nabla\omega(\mathbf{u}) - \nabla\omega(\mathbf{v})\|_2 \le P\|\mathbf{u} - \mathbf{v}\|_2^{\tilde{\alpha}}.$$

**Remark 6.** It can be established that Assumption 3 leads to the Lipschitz continuity of the activation function when $\tilde{\alpha} = 0$. Furthermore, $\tilde{\alpha} = 1$ implies the smoothness of the activation function. As a result, Assumption 3 stands as notably milder in comparison to the assumption found in prior works [42, 9], which mandates the activation function's smoothness. In order to facilitate analysis without introducing a significant disparity between theory and practical application, we often use modified ReLU function

$$\omega(x) = \begin{cases} 0, x \le 0, \\ x^q, 0 < x \le \left(\frac{1}{q}\right)^{\frac{1}{q-1}}, \\ x - \left(\frac{1}{q}\right)^{\frac{1}{q-1}} + \left(\frac{1}{q}\right)^{\frac{q}{q-1}}, x > \left(\frac{1}{q}\right)^{\frac{1}{q-1}}. \end{cases}$$

This modified function, controlled by the hyperparameter $q \in (1, 2]$, not only satisfies Assumption 3 but also maintains an acceptable approximation to the vanilla ReLU function.

**Lemma 1** (Proposition 4.1 in [37])**.** *Suppose that Assumption 2 and 3 hold. Denote by $\mathcal{F}$ a specific GNN, for any $\mathbf{w}, \mathbf{w}' \in \mathcal{W}$ and $\mathbf{x} \in \boldsymbol{X}_N$, the objective $\ell(\mathbf{w}; \mathbf{x})$ satisfies*

$$|\ell(\mathbf{w}; \mathbf{x}) - \ell(\mathbf{w}'; \mathbf{x})| \le L_{\mathcal{F}}\|\mathbf{w} - \mathbf{w}'\|_2,$$

*and*

$$\|\nabla\ell(\mathbf{w}; \mathbf{x}) - \nabla\ell(\mathbf{w}'; \mathbf{x})\| \le P_{\mathcal{F}}\max\{\|\mathbf{w} - \mathbf{w}'\|_2^{\tilde{\alpha}}, \|\mathbf{w} - \mathbf{w}'\|_2\},$$

*with constant $L_{\mathcal{F}}$ and $P_{\mathcal{F}}$.*

**Remark 7.** [37] demonstrates that several widely used structured networks in GNNs such as GCN [20], GCNII [7], SGC [45], APPNP [17] and GPR-GNN [8] satisfy Lemma 1. We leverage the properties of these network structures in Lemma 1 to derive improved upper bounds using our concentration inequalities instead of [13].

The following two assumptions are introduced to obtain the optimization error.

**Assumption 4.** *Assume that there exist a constant $G > 0$ such that for all $\mathbf{x} \in \mathbf{Z}$*

$$\sqrt{\eta_t}\|\nabla\ell(\mathbf{w}_t; \mathbf{x})\|_2 \leq G$$

*holds for any $t \in \mathbb{N}$, where $\{\eta_t\}_{t=1}^T$ is learning rates.*

**Assumption 5.** *Assume that there exists a constant $\sigma_0 > 0$ such that for $\forall t \in \mathbb{N}_+$, the following inequality holds*

$$\mathbb{E}_{jt}[\|\nabla\ell(\mathbf{w}; \mathbf{x}_{jt})\|^2] \leq \sigma_0^2.$$

**Remark 8.** Assumption 4 [26, 27] requires a bound on the product of the gradient and the square root of the step sizes. This condition is weaker than the commonly employed bounded gradient assumption [18, 21], as the learning rate naturally approaches zero throughout the iteration process. Assumption 5 requires the boundness of variances of stochastic gradients, which is a standard assumption in stochastic optimization studies [21, 26, 27].

Now, we can derive the risk bounds of GNNs with SGD.

**Theorem 5.** *Suppose Assumptions 2, 3, 4, and 5 hold, and assume the objective function $\ell(\cdot; \cdot)$ be sub-Gaussian. Suppose that the step sizes $\{\eta_t\}$ satisfies $\eta_t = \frac{1}{t+t_0}$ such that $t_0 \geq \max\{(2P)^{1/\alpha}, 1\}$. For any $\delta \in (0, 1)$, with probability $1 - \delta$,*

*(a). If $\alpha \in (0, \frac{1}{2})$, we have*

$$R_u(\mathbf{w}_1^{(T+1)}) - \hat{R}_m(\mathbf{w}^{(T+1)}) = \mathcal{O}\left(L_{\mathcal{F}}\frac{\sqrt{N}}{u}\log^{\frac{1}{2}}(T)T^{\frac{1-2\alpha}{2}}\log\left(\frac{1}{\delta}\right) + \frac{N\log\left(\frac{1}{\delta}\right)}{u\sqrt{m}}\right).$$

*(b). If $\alpha = \frac{1}{2}$, we have*

$$R_u(\mathbf{w}^{(T+1)}) - \hat{R}_m(\mathbf{w}^{(T+1)}) = \mathcal{O}\left(L_{\mathcal{F}}\frac{\sqrt{N}}{u}\log(T)\log\left(\frac{1}{\delta}\right) + \frac{N\log\left(\frac{1}{\delta}\right)}{u\sqrt{m}}\right).$$

*(c). If $\alpha \in (\frac{1}{2}, 1]$, we have*

$$R_u(\mathbf{w}^{(T+1)}) - \hat{R}_m(\mathbf{w}^{(T+1)}) = \mathcal{O}\left(L_{\mathcal{F}}\frac{\sqrt{N}}{u}\log^{\frac{1}{2}}(T)\log\left(\frac{1}{\delta}\right) + \frac{N\log\left(\frac{1}{\delta}\right)}{u\sqrt{m}}\right).$$

**Remark 9.** Similar result for sub-exponential loss functions is given in Appendix C. Generally, comparing our bound with [37], their bound is of order $\mathcal{O}\left(\left(\frac{1}{m} + \frac{1}{u}\right)\sqrt{m+u}\right)$ after the $L_{\mathcal{F}}$ but our bounds are of order $\mathcal{O}\left(\frac{\sqrt{m+u}}{u}\right)$, at the same time, we have an extra term $\frac{m+u}{u\sqrt{m}}$, which is introduced due to the variance information. Notice that it's not as if they didn't have the second term, because their first term is larger than the second one and so the final magnitude doesn't change. Our results are better when $m = o(N^{2/5})$. We can take a more visual example to demonstrate the advantages of our bounds. For $m = \Theta(N^{1/5})$, our bound is of order $\mathcal{O}\left(\frac{1}{\sqrt{m}}\right)$ but their bound is of order $\mathcal{O}\left(m^3\right)$, which fails to provide a reasonable generalization guarantee.

Similarly, we can also derive a upper bound of the test error under PL condition following proof trajectory of [37].

**Assumption 6** (PL-condition). *Suppose that there exists a constant $\mu$ such that for all $\mathbf{w} \in \mathcal{W}$,*

$$\hat{R}_m(\mathbf{w}) - \hat{R}_m(\hat{\mathbf{w}}^*) \leq \frac{1}{2\mu}\|\nabla\hat{R}_m(\mathbf{w})\|_2,$$

*holds for the given set $\mathbf{X}_m$ from $\mathbf{X}_N$.*

**Corollary 1.** *Suppose Assumptions 2, 3, 4, and 5 hold and assume the objective function $\ell(\cdot;\cdot)$ be sub-Gaussian. Suppose that the learning rate $\{\eta_t\}$ satisfies $\eta_t = \frac{2}{\mu(t+t_0)}$ such that $t_0 \geq \max\{\frac{2}{\mu}(2P)^{\frac{1}{\alpha}}, 1\}$. For any $\delta \in (0,1)$, with probability $1-\delta$,*

*(a). If $\alpha \in (0, \frac{1}{2})$, we have*

$$R_u(\mathbf{w}^{(T+1)}) - \hat{R}_m(\mathbf{w}^*) = \mathcal{O}\left( L_{\mathcal{F}} \frac{\sqrt{N}}{u} \log^{\frac{1}{2}}(T) T^{\frac{1}{2}-\alpha} \log\left(\frac{1}{\delta}\right) + \frac{N \log\left(\frac{1}{\delta}\right)}{u\sqrt{m}} + \frac{1}{T^\alpha} \right),$$

*(b). If $\alpha = \frac{1}{2}$, we have*

$$R_u(\mathbf{w}^{(T+1)}) - \hat{R}_m(\mathbf{w}^*) = \mathcal{O}\left( L_{\mathcal{F}} \frac{\sqrt{N}}{u} \log(T) \log\left(\frac{1}{\delta}\right) + \frac{N \log\left(\frac{1}{\delta}\right)}{u\sqrt{m}} + \frac{1}{T^\alpha} \right).$$

*(c). If $\alpha \in (\frac{1}{2}, 1)$, we have*

$$R_u(\mathbf{w}^{(T+1)}) - \hat{R}_m(\mathbf{w}^*) = \mathcal{O}\left( L_{\mathcal{F}} \frac{\sqrt{N}}{u} \log^{\frac{1}{2}}(T) \log(1/\delta) + \frac{N \log\left(\frac{1}{\delta}\right)}{u\sqrt{m}} + \frac{1}{T^\alpha} \right).$$

*(d). If $\alpha = 1$, we have*

$$R_u(\mathbf{w}^{(T+1)}) - R_u(\mathbf{w}^*) = \mathcal{O}\left( L_{\mathcal{F}} \frac{\sqrt{N}}{u} \log^{\frac{1}{2}}(T) \log(1/\delta) + \frac{N \log\left(\frac{1}{\delta}\right)}{u\sqrt{m}} + \frac{\log(T) \log^3(1/\delta)}{T} \right).$$

**Remark 10.** For completeness, we present Corollary 1 for sub-Gaussian and Corollary 2 (See Appendix C.2) for sub-exponential. There is nothing special about the proofs, which simply combine Theorem 5 and Theorem 11 with existing optimization results. The results under the sub-exponential distribution are provided in Appendix 4.2. It is worth point out that all the popular neural network structures introduced in [37] can be applied to our results to obtain bounds that make sense.

Our work in this section differs significantly from that of [37]. They used the concentration inequalities based on [13] to derive generalization bounds, while proving that certain modern neural network structures satisfy Lipschitz continuity under their assumptions. In contrast, we employ newly proposed concentration inequalities that relax the boundness condition and also consider variance information which obtain improved rates under the same settings.

While previous papers have utilized technologies based on concentration inequalities proposed by [13] and then bound the transductive Rademacher complexity, deriving the generalization error using our new inequality is not straightforward. We need to derive the upper bounds for $\sigma_{\mathbf{w}}^2$, $E_m$, and $\|\max_{\mathbf{x}} \sup_{f \in \mathcal{F}_{\mathbf{w}}} f_{\mathbf{w}}(\mathbf{x})\|_{\psi_\alpha}^2$, respectively. $\sigma_{\mathbf{w}}^2$ needs to be bounded using concentration inequalities for unbounded distributions. For the sub-exponential distribution, we even need to introduce the concentration inequalities under the sub-Weibull distribution to address the issue. $E_m$ is introduced due to the Hoeffding's reduction method and is distinct from the traditional gap between the population and the samples. This requires us to convert it into Rademacher complexity and then use the covering number to obtain the upper bound. The term $\|\max_{\mathbf{x}} \sup_{f \in \mathcal{F}_{\mathbf{w}}} f_{\mathbf{w}}(\mathbf{x})\|_{\psi_\alpha}^2$ is introduced due to the unbounded assumption. We utilize pisier's inequality [34] to present the $\max$ operator before the Orlicz norm.

## 5 CONCLUSION

In this paper, we focus on transductive learning settings. Firstly, we introduce two newly concentration inequalities for the suprema of empirical processes sampled without replacement for unbounded functions. Using our inequalities, we derive the first fast risk bounds for ERM in transductive learning under bounded losses. On the other hand, we provide improved risk bounds for GNNs with SGD, which is better than the state-of-the-art work [37] when $m = o(N^{2/5})$.

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

## A   ADDITIONAL DEFINITIONS AND LEMMATA

**Theorem 6** ([19])**.** *Let $\{U_1, \ldots, U_m\}$ and $\{W_1, \ldots, W_m\}$ be sampled uniformly from a finite set of d-dimensional vectors $\{\boldsymbol{v}_1, \ldots, \boldsymbol{v}_N\} \subset \mathbb{R}^d$ with and without replacement, respectively. Then, for any continuous and convex function $F : \mathbb{R}^d \to \mathbb{R}$, the following holds:*

$$\mathbb{E}\left[F\left(\sum_{i=1}^m W_i\right)\right] \leq \mathbb{E}\left[F\left(\sum_{i=1}^m U_i\right)\right].$$

**Lemma 2** ([38])**.** *Let $\boldsymbol{x} = (x_1, \ldots, x_d)^T \in \mathbb{R}^d$. Then the following function is convex for all $\lambda > 0$*

$$F(\boldsymbol{x}) = \exp\left(\lambda \sup_{i=1,\ldots,d} x_i\right).$$

**Theorem 7** (Theorem 4 via Pisier's inequality [34])**.** *For independent real random variables $Y_i, \ldots, Y_n$, we have the following inequality that*

$$\left\|\max_{i \leq n} Y_i\right\|_{\psi_\alpha} \leq K_\alpha \max_{i \leq n} \|Y_i\|_{\psi_\alpha} \log^{1/\alpha} n,$$

*where $K_\alpha$ is a positive constant.*

**Definition 4** (Rademacher complexity [44])**.** *For a function class $\mathcal{F}$ that consists of mappings from $\mathcal{Z}$ to $\mathbb{R}$, define*

$$\mathfrak{R}\mathcal{F} := \mathbb{E}_{\mathbf{x},v} \sup_{f \in \mathcal{F}} \frac{1}{n} \sum_{i=1}^n v_i f(\mathbf{x}_i) \quad and \quad \mathfrak{R}_n\mathcal{F} := \mathbb{E}_v \sup_{f \in \mathcal{F}} \frac{1}{n} \sum_{i=1}^n v_i f(\mathbf{x}_i),$$

*as the Rademacher complexity and the empirical Rademacher complexity of $\mathcal{F}$, respectively, where $\{v_i\}_{i=1}^n$ are i.i.d. Rademacher variables for which $\mathbb{P}(v_i = 1) = \mathbb{P}(v_i = -1) = \frac{1}{2}$.*

**Definition 5** (Covering number [44])**.** *Assume $(\mathcal{M}, \mathrm{metr}(\cdot, \cdot))$ is a metric space, and $\mathcal{F} \subseteq \mathcal{M}$. The $\varepsilon$-convering number of the set $\mathcal{F}$ with respect to a metric $\mathrm{metr}(\cdot, \cdot)$ is the size of its smallest $\varepsilon$-net cover:*

$$\mathcal{N}(\varepsilon, \mathcal{F}, \mathrm{metr}) = \min\{m : \exists f_1, \ldots, f_m \in \mathcal{F} \text{ such that } \mathcal{F} \subseteq \cup_{j=1}^m \mathcal{B}(f_j, \varepsilon)\},$$

*where $\mathcal{B}(f, \varepsilon) := \{\tilde{f} : \mathrm{metr}(\tilde{f}, f) \leq \varepsilon\}$.*

**Lemma 3** (Dudley's integral bound [35])**.** *Given $r > 0$ and class $\mathcal{F}$ that consists of functions defined on $\mathcal{Z}$,*

$$\mathfrak{R}_n\{f \in \mathcal{F} : \mathbb{P}_n[f^2] \leq r\} \leq \inf_{\varepsilon_0 > 0} \left\{4\varepsilon_0 + 12 \int_{\varepsilon_0}^{\sqrt{r}} \sqrt{\frac{\log \mathcal{N}(\varepsilon, \mathcal{F}, L_2(\mathbb{P}_n))}{n}} d\varepsilon\right\}.$$

**Definition 6** ([43])**.** *A random variable $X$ is sub-Weibull random variables with taill parameter $\theta$ when for any $x > 0$,*

$$\mathbb{P}(X \geq x) = \exp(-bx^{1/\theta}), \text{ for some } b > 0, \theta > 0.$$

**Lemma 4. (Concentration of the sum for sub-Weibull distribution [43])** Let that $X_1, \ldots, X_n$ be identically distributed sub-Weibull random variables with tail parameter $\theta$. Then, for all $x \geq nK_\theta$, we have

$$\mathbb{P}\left(\left|\sum_{i=1}^n X_i\right| \geq x\right) \leq \exp\left(-\left(\frac{x}{nK_\theta}\right)^{1/\theta}\right),$$

for some constant $K_\theta$ dependent on $\theta$.

**Theorem 8** ([1])**.** *Let $X_1, \ldots, X_m$ be independent random variables with values in a measurable space $(\mathbb{S}, \mathbb{B})$ and let $\mathcal{F}$ be a countable class of measurable functions $f : \mathcal{S} \to [-a, a]$, such that for all $i$, $\mathbb{E}f(X_i) = 0$. Consider the random variable*

$$Q = \sup_{f \in \mathcal{F}} \sum_{i=1}^m f(X_i)$$

*and*

$$\sigma^2 = \sup_{f \in \mathcal{F}} \mathbb{E} f(X_1)^2.$$

*Then, for all $0 < \eta \leq 1$, $\beta > 0$ there exists a constant $C = C(\eta, \beta)$, such that for all $t > 0$,*

$$\mathbb{P}(Q - (1 + \eta)\mathbb{E}Q \geq t) \leq \exp\left(-\frac{t^2}{2(1 + \beta)m\sigma^2}\right) + \exp\left(-\frac{t}{Ca}\right),$$

*and*

$$\mathbb{P}(Q - (1 - \eta)\mathbb{E}Q \leq -t) \leq \exp\left(-\frac{t^2}{2(1 + \beta)m\sigma^2}\right) + \exp\left(-\frac{t}{Ca}\right).$$

**Theorem 9. (Tail inequality for suprema of empirical process corresponding to classes of sub-Gaussian functions)** Let $X_1, \ldots, X_m$ be independent random variables with values in a measurable space $(\mathbb{S}, \mathbb{B})$ and let $\mathcal{F}$ be a countable class of measurable functions $f : \mathcal{S} \to \mathbb{R}$. Assume that for every $f \in \mathcal{F}$ and every $i$, $\mathbb{E} f(X_i) = 0$ and $\|\sup_f |f(X_i)|\|_{\psi_2} < \infty$. Let

$$Q = \sup_{f \in \mathcal{F}} \sum_{i=1}^{m} f(X_i)$$

and

$$\sigma^2 = \sup_{f \in \mathcal{F}} \mathbb{E} f(X_i)^2.$$

Then, for all $0 < \eta < 1$ and $\beta > 0$, there exists a constant $C = C(\eta, \beta)$, such that for all $epsilon > 0$,

$$\mathbb{P}(Q - (1 + \eta)\mathbb{E}Q \geq t) \leq \exp\left(-\frac{t^2}{2(1 + \beta)m\sigma^2}\right) + 3\exp\left(-\left(\frac{t}{C\|\max_i \sup_{f \in \mathcal{F}} f(X_i)\|_{\psi_2}}\right)^2\right),$$

and

$$\mathbb{P}(Q - (1 - \eta)\mathbb{E}Q \leq -t) \leq \exp\left(-\frac{t^2}{2(1 + \beta)m\sigma^2}\right) + 3\exp\left(-\left(\frac{t}{C\|\max_i \sup_{f \in \mathcal{F}} f(X_i)\|_{\psi_2}}\right)^2\right).$$

**Theorem 10. (Tail inequality for suprema of empirical process corresponding to classes of sub-exponential functions)** Let $X_1, \ldots, X_m$ be independent random variables with values in a measurable space $(\mathbb{S}, \mathbb{B})$ and let $\mathcal{F}$ be a countable class of measurable functions $f : \mathcal{S} \to \mathbb{R}$. Assume that for every $f \in \mathcal{F}$ and every $i$, $\mathbb{E} f(X_i) = 0$ and $\|\sup_f |f(X_i)|\|_{\psi_1} < \infty$. Let

$$Q = \sup_{f \in \mathcal{F}} \sum_{i=1}^{m} f(X_i)$$

and

$$\sigma^2 = \sup_{f \in \mathcal{F}} \mathbb{E} f(X_i)^2.$$

Then, for all $0 < \eta < 1$ and $\beta > 0$, there exists a constant $C = C(\eta, \beta)$, such that for all $epsilon > 0$,

$$\mathbb{P}(Q - (1 + \eta)\mathbb{E}Q \geq t) \leq \exp\left(-\frac{t^2}{2(1 + \beta)m\sigma^2}\right) + 3\exp\left(-\frac{t}{C\|\max_i \sup_{f \in \mathcal{F}} f(X_i)\|_{\psi_1}}\right),$$

and

$$\mathbb{P}(Q - (1 - \eta)\mathbb{E}Q \leq -t) \leq \exp\left(-\frac{t^2}{2(1 + \beta)m\sigma^2}\right) + 3\exp\left(-\frac{t}{C\|\max_i \sup_{f \in \mathcal{F}} f(X_i)\|_{\psi_1}}\right).$$

The proofs of Theorem 9 and Theorem 10 are similar with [1], which under the assumption that the summands have finite $\psi_\alpha$ Orlicz norm with $\alpha \in (0, 1)$ and they analyze the random variable $Q = \sup_{f \in \mathcal{F}} |\sum_{i=1}^{m} f(X_i)|$. However, in this paper, we consider $Q = \sup_{f \in \mathcal{F}} \sum_{i=1}^{m} f(X_i)$. In consequence we give the sub-gaussian and sub-exponential version ($\alpha = 1, 2$) for the sake of completeness here.

*Proof of Theorem 9 and Theorem 10.* Without loss of generality, we assume that

$$t / \|\max_{1 \le i \le m} \sup_{f \in \mathcal{F}} f(X_i)\|_{\psi_\alpha} > K(\alpha, \eta, \beta), \tag{1}$$

otherwise we can make the theorem trivial by choosing the constant $C = C(\alpha, \eta, \beta)$ to be large enough. The conditions on the constant $K(\alpha, \eta, \beta)$ will be imposed later in the following proof.

Let $\varepsilon = \varepsilon(\beta) > 0$ which will be determined later and for all $f \in \mathcal{F}$ consider the truncated functions $f_1(x) = f(x) \mathbf{1}_{\{\sup_{f \in \mathcal{F}} |f(x)| \le \rho\}}$ (the truncation level $\rho$ will be determined and fixed later). Define the functions $f_2(x) = f(x) - f_1(x) = f(x) \mathbf{1}_{\{\sup_{f \in \mathcal{F}} |f(x)| > \rho\}}$. Let $\mathcal{F}_i = \{f_i : f \in \mathcal{F}\}$. Then we have

$$Q = \sup_{f \in \mathcal{F}} \sum_{i=1}^m f(X_i) \le \sup_{f_1 \in \mathcal{F}_1} \sum_{i=1}^m (f_1(X_i) - \mathbb{E}f_1(X_i)) + \sup_{f_2 \in \mathcal{F}_2} \sum_{i=1}^m (f_2(X_i) - \mathbb{E}f_2(X_i)), \tag{2}$$

and

$$Q \ge \sup_{f_1 \in \mathcal{F}_1} \sum_{i=1}^m (f_1(X_i) - \mathbb{E}f_1(X_i)) - \sup_{f_2 \in \mathcal{F}_2} \sum_{i=1}^m (f_2(X_i) - \mathbb{E}f_2(X_i)), \tag{3}$$

where the above inequalities satisfy because of the fact that $\mathbb{E}f_1(X_i) + \mathbb{E}f_2(X_i) = 0$ for all $f \in \mathcal{F}$. Similarly, by Jensen's inequality, we have

$$\begin{aligned} & \mathbb{E} \sup_{f_1 \in \mathcal{F}_1} \sum_{i=1}^m (f_1(X_i) - \mathbb{E}f_1(X_i)) - 2\mathbb{E} \sup_{f_2 \in \mathcal{F}_2} \sum_{i=1}^m f_2(X_i) \\ \le & \mathbb{E}Q \\ \le & \sup_{f_1 \in \mathcal{F}_1} \sum_{i=1}^m (f_1(X_i) - \mathbb{E}f_1(X_i)) + 2\mathbb{E} \sup_{f_2 \in \mathcal{F}_2} \sum_{i=1}^m f_2(X_i). \end{aligned} \tag{4}$$

Denoting

$$A = \mathbb{E} \sup_{f_1 \in \mathcal{F}_1} \sum_{i=1}^m (f_1(X_i) - \mathbb{E}f_1(X_i))$$

and

$$B = \mathbb{E} \sup_{f_2 \in \mathcal{F}_2} \sum_{i=1}^m f_2(X_i).$$

Combining (2) and (4), we get

$$\begin{aligned} & \mathbb{P}(Q - (1+\eta)\mathbb{E}Q \ge t) \\ \le & \mathbb{P}\left( \sup_{f_1 \in \mathcal{F}_1} \sum_{i=1}^m (f_1(X_i) - \mathbb{E}f_1(X_i)) \ge (1+\eta)\mathbb{E}Q + (1-\varepsilon)t \right) \\ & + \mathbb{P}\left( \sup_{f_2 \in \mathcal{F}_2} \sum_{i=1}^m (f_2(X_i) - \mathbb{E}f_2(X_i)) \ge \varepsilon t \right) \\ \le & \mathbb{P}\left( \sup_{f_1 \in \mathcal{F}_1} \sum_{i=1}^m (f_1(X_i) - \mathbb{E}f_1(X_i)) \ge (1+\eta)A - 4B + (1-\varepsilon)t \right) \\ & + \mathbb{P}\left( \sup_{f_2 \in \mathcal{F}_2} \sum_{i=1}^m (f_2(X_i) - \mathbb{E}f_2(X_i)) \ge \varepsilon t \right). \end{aligned} \tag{5}$$

Similarly, combing (3) and (4), we have

$$
\mathbb{P}(Q - (1-\eta)\mathbb{E}Q \le -t)
$$

$$
\le \mathbb{P}\left( \sup_{f_1 \in \mathcal{F}_1} \sum_{i=1}^{m}(f_1(X_i) - \mathbb{E}f_1(X_i)) \le (1-\eta)\mathbb{E}Q - (1-\varepsilon)t \right)
$$

$$
+ \mathbb{P}\left( \sup_{f_2 \in \mathcal{F}_2} \sum_{i=1}^{m}(f_2(X_i) - \mathbb{E}f_2(X_i)) \ge \varepsilon t \right) \tag{6}
$$

$$
\le \mathbb{P}\left( \sup_{f_1 \in \mathcal{F}_1} \sum_{i=1}^{m}(f_1(X_i) - \mathbb{E}f_1(X_i)) \ge (1-\eta)A + 2B - (1-\varepsilon)t \right)
$$

$$
+ \mathbb{P}\left( \sup_{f_2 \in \mathcal{F}_2} \sum_{i=1}^{m}(f_2(X_i) - \mathbb{E}f_2(X_i)) \ge \varepsilon t \right).
$$

Next, we need to choose proper truncation level $\rho$ in a way, which would allow to bound the first summands on the right-hand sides of (5) and (6) with Theorem 8.

Let us set

$$
\rho = 8\mathbb{E}\max_{1 \le i \le m}\sup_{f \in \mathcal{F}} f(X_i) \le K_\alpha \left\| \max_{1 \le i \le m}\sup_{f \in \mathcal{F}} f(X_i) \right\|_{\psi_\alpha}. \tag{7}
$$

Notice that by the Chebyshev inequality and the definition of the class $\mathcal{F}_2$, we have

$$
\mathbb{P}\left( \max_{k \le m}\sup_{f \in \mathcal{F}} \sum_{i=0}^{k} f_2(X_i) > 0 \right) \le \mathbb{P}\left( \max_i \sup_f f(X_i) > \rho \right) \le 1/8.
$$

Thus by the Hoffmann-Jorgensen inequality [25], we get

$$
B = \mathbb{E}\sup_{f_2 \in \mathcal{F}_2} \sum_{i=1}^{m} f_2(X_i) \le 8\mathbb{E}\max_{1 \le i \le m}\sup_{f \in \mathcal{F}} f(X_i). \tag{8}
$$

In consequence

$$
\mathbb{E}\sup_{f_2 \in \mathcal{F}_2}\sum_{i=1}^{m}(f_2(X_i) - \mathbb{E}f_2(X_i)) \le 16\mathbb{E}\max_{1 \le i \le m}\sup_{f \in \mathcal{F}} f(X_i) \le K_\alpha \left\| \max_{1 \le i \le m}\sup_{f \in \mathcal{F}} f(X_i) \right\|_{\psi_\alpha}.
$$

Thus, we have

$$
\left\| \max_{1 \le i \le m}\sup_{f \in \mathcal{F}} f_2(X_i) - \mathbb{E}f_2(X_i) \right\|_{\psi_\alpha} \le \left\| \max_{1 \le i \le m}\sup_{f \in \mathcal{F}} f_2(X_i) \right\|_{\psi_\alpha} + \left\| \mathbb{E}\max_{1 \le i \le m}\sup_{f \in \mathcal{F}} f_2(X_i) \right\|_{\psi_\alpha}
$$

$$
\le 2\left\| \max_{1 \le i \le m}\sup_{f \in \mathcal{F}} f_2(X_i) \right\|_{\psi_\alpha}
$$

$$
\le 2\left\| \max_{1 \le i \le m}\sup_{f \in \mathcal{F}} f(X_i) \right\|_{\psi_\alpha},
$$

where the above inequality holds because $\|\cdot\|_{\psi_\alpha}$ ($\alpha = 1, 2$) is a standard norm. Then, by Theorem 6.21 of [25], we obtain

$$
\left\| \sup_{f_2 \in \mathcal{F}_2}\sum_{i=1}^{m}(f_2(X_i) - \mathbb{E}f_2(X_i)) \right\|_{\psi_\alpha} \le K_\alpha \left\| \max_{1 \le i \le m}\sup_{f \in \mathcal{F}} f(X_i) \right\|_{\psi_\alpha},
$$

which implies

$$\mathbb{P}\left(\sup_{f_2 \in \mathcal{F}_2} \sum_{i=1}^{m} f_2(X_i) - \mathbb{E}f_2(X_i) \geq \varepsilon t\right) \leq 2\exp\left(-\left(\frac{\varepsilon t}{K\left\|\max_{1\leq i \leq n} \sup_{f \in \mathcal{F}} f(X_i)\right\|_{\psi_\alpha}}\right)^2\right). \quad (9)$$

Next, let us choose $\varepsilon < 1/10$ and such that

$$(1 - 5\varepsilon)^{-2}(1 + \beta/2) \leq (1 + \beta). \quad (10)$$

Since $\varepsilon$ is a function of $\beta$, in view of (7) and (8), we can choose the constant $K(\alpha, \eta, \beta)$ in (1) to be large enough to assure that

$$B \leq 8\mathbb{E} \max_{1\leq i \leq m} \sup_{f \in \mathcal{F}} f(X_i) \leq \varepsilon t.$$

Notice that for every $f \in \mathcal{F}$, we have $\mathbb{E}(f_1(X_i) - \mathbb{E}f_1(X_i))^2 \leq \mathbb{E}f_1(X_i)^2 \leq \mathbb{E}f(X_i)^2$.

Thus, using inequalities (5), (6), (9) and Theorem 8 (applied for $\eta$ and $\beta/2$), we obtain

$$\mathbb{P}(Q - (1+\eta)\mathbb{E}Q \geq t), \quad \mathbb{P}(Q - (1-\eta)\mathbb{E}Q \leq -t)$$
$$\leq \exp\left(-\frac{t^2(1-5\varepsilon)^2}{2(1+\beta/2)m\sigma^2}\right) + \exp\left(-\frac{(1-5\varepsilon)t}{K(\alpha, \eta, \beta)\rho}\right)$$
$$+ 2\exp\left(-\left(\frac{\varepsilon t}{K_\alpha \|\max_{1\leq i \leq m} \sup_{f\in \mathcal{F}} f(X_i)\|_{\psi_\alpha}}\right)^\alpha\right).$$

Since $\varepsilon < 1/10$, using (7) we can see that for all $t$ with $K(\alpha, \eta, \beta)$ large enough, we have

$$\exp\left(-\frac{(1-5\varepsilon)t}{K(\alpha, \eta, \beta)\rho}\right), \exp\left(-\left(\frac{\varepsilon t}{K_\alpha \|\max_{1\leq i \leq m} \sup_{f\in \mathcal{F}} f(X_i)\|_{\psi_\alpha}}\right)^\alpha\right)$$
$$\leq \exp\left(-\left(\frac{t}{\widetilde{C}(\alpha, \eta, \beta)\|\max_{1\leq i \leq m} \sup_{f\in \mathcal{F}} f(X_i)\|_{\psi_\alpha}}\right)^\alpha\right).$$

Therefore, for all t,

$$\mathbb{P}(Q - (1+\eta)\mathbb{E}Q \geq t), \quad \mathbb{P}(Q - (1-\eta)\mathbb{E}Q \leq -t)$$
$$\leq \exp\left(-\frac{t^2(1-5\varepsilon)^2}{2(1+\beta/2)m\sigma^2}\right) + 3\exp\left(-\left(\frac{t}{\widetilde{C}(\alpha, \eta, \beta)\|\max_{1\leq i \leq m} \sup_{f\in \mathcal{F}} f(X_i)\|_{\psi_\alpha}}\right)^\alpha\right).$$

Finally, we use (10) to finish the proof.

$\square$

**Lemma 5. (Moment-generating function inequality for suprema of empirical process corresponding to classes of sub-Gaussian functions)** Let $X$ and $Q$ be defined in Theorem 9, then for all $0 < \eta < 1$ and $\beta > 0$, there exists a constant $C = C(\eta, \beta)$, such that

$$\mathbb{E}\exp(\lambda(Q - (1+\eta)\mathbb{E}Q)) \leq \exp\left(4(1+\beta)m\sigma^2\lambda^2\right) + 3\exp\left(2\left(C\lambda\left\|\max_i \sup_{f\in \mathcal{F}} f(X_i)\right\|_{\psi_2}\right)^2\right).$$

**Lemma 6. (Moment-generating function inequality for suprema of empirical process corresponding to classes of sub-exponential functions)** Let $X$ and $Q$ be defined in Theorem 10, then for all $0 < \eta < 1$ and $\beta > 0$, there exists a constant $C = C(\eta, \beta)$, such that

$$\mathbb{E}\exp(\lambda(Q - (1+\eta)\mathbb{E}Q)) \leq \exp\left(4(1+\beta)m\sigma^2\lambda^2\right) + \exp\left(12\left(C\lambda\left\|\max_i \sup_{f\in \mathcal{F}} f(X_i)\right\|_{\psi_1}\right)^2\right).$$

*Proof of Lemma 5.* In the proof we use the notation $\lesssim$ between two positive sequences $(a_k)_k$ and $(b_k)_k$, writing $a_k \leq b_k$, if there exists a constant $C > 0$ such that for all integer $k$, $a_k \leq Cb_k$.

According to Theorem 9, we have

$$\mathbb{P}(|Q - (1+\eta)\mathbb{E}Q| \geq t) \leq 2\exp\left(-\frac{t^2}{2(1+\beta)m\sigma^2}\right) + 6\exp\left(-\frac{t^2}{C^2\|\max_i \sup_{f\in\mathcal{F}} |f(X_i)|\|_{\psi_2}^2}\right).$$

Let the random variable $Y = Q - (1+\eta)\mathbb{E}Q$ we have that for any $k \geq 1$,

$$\mathbb{E}[|Y|^k]$$
$$= \int_0^\infty \mathbb{P}\left(|Y|^k > t\right) dt$$
$$= \int_0^\infty \mathbb{P}\left(|Y| > t^{1/k}\right) dt$$
$$\leq \int_0^\infty 2\exp\left(-\frac{t^{2/k}}{2(1+\beta)m\sigma^2}\right) dt + \int_0^\infty 6\exp\left(-\frac{t^{2/k}}{C^2\|\max_i \sup_{f\in\mathcal{F}} f(X_i)\|_{\psi_2}^2}\right) dt$$
$$= \left(2(1+\beta)m\sigma^2\right)^{k/2} k \int_0^\infty e^{-u} u^{k/2-1} du + 3k\left(C\left\|\max_i \sup_{f\in\mathcal{F}} f(X_i)\right\|_{\psi_2}\right)^k \int_0^\infty e^{-v} v^{k/2-1} dv$$
$$= \left(2(1+\beta)m\sigma^2\right)^{k/2} k\Gamma(k/2) + 3k\left(C\left\|\max_i \sup_{f\in\mathcal{F}} f(X_i)\right\|_{\psi_2}\right)^k \Gamma(k/2),$$

where we denote $u = \frac{t^{2/k}}{2(1+\beta)m\sigma^2}$ and $v = \frac{t^{2/k}}{C^2\|\max_i \sup_{f\in\mathcal{F}} |f(X_i)|\|_{\psi_2}^2}$ in the third equality.

Next, we use the Taylor expansion of the exponential function as follows. For $\lambda > 0$, we have

$$\mathbb{E}\exp(\lambda Y)$$
$$= 1 + \sum_{k=2}^\infty \frac{\lambda^k \mathbb{E}[|Y|^k]}{k!}$$
$$\lesssim 1 + \sum_{k=2}^\infty \frac{\left(2(1+\beta)m\sigma^2\lambda^2\right)^{k/2} k\Gamma(k/2) + 3k(C\lambda\|\max_i \sup_{f\in\mathcal{F}} f(X_i)\|_{\psi_2})^k \Gamma(k/2)}{k!}$$
$$= 1 + \sum_{k=1}^\infty \frac{\left(2(1+\beta)m\sigma^2\lambda^2\right)^k 2k\Gamma(k)}{(2k)!} + \sum_{k=1}^\infty \frac{\left(2(1+\beta)m\sigma^2\lambda^2\right)^{k+1/2} (2k+1)\Gamma(k+1/2)}{(2k+1)!}$$
$$+ \sum_{k=1}^\infty \frac{6k(C\lambda\|\max_i \sup_{f\in\mathcal{F}} f(X_i)\|_{\psi_2})^{2k}\Gamma(k)}{k!}$$
$$+ \sum_{k=1}^\infty \frac{3(2k+1)(C\lambda\|\max_i \sup_{f\in\mathcal{F}} f(X_i)\|_{\psi_2})^{2k+1}\Gamma(k+1/2)}{k!}$$
$$\leq 1 + \left(2 + \sqrt{2(1+\beta)m\sigma^2\lambda^2}\right) \sum_{k=1}^\infty \frac{\left(2(1+\beta)m\sigma^2\lambda^2\right)^k k!}{(2k)!}$$
$$+ \left(6 + C\lambda\left\|\max_i \sup_{f\in\mathcal{F}} f(X_i)\right\|_{\psi_2}\right) \sum_{k=1}^\infty \frac{(C\lambda\|\max_i \sup_{f\in\mathcal{F}} f(X_i)\|_{\psi_2})^{2k} k!}{(2k)!},$$

where the second equality satisfies because of commutative property of positive convergent series. This implies that

$$\mathbb{E}\exp(\lambda Y)$$

$$\lesssim 1 + \left(1 + \sqrt{\frac{(1+\beta)m\sigma^2\lambda^2}{2}}\right)\sum_{k=1}^{\infty}\frac{\left(2(1+\beta)m\sigma^2\lambda^2\right)^k}{(2k)!}$$

$$+ \left(3 + \frac{C\lambda\left\|\max_i\sup_{f\in\mathcal{F}}f(X_i)\right\|_{\psi_2}}{2}\right)\sum_{k=1}^{\infty}\frac{(C\lambda\|\max_i\sup_{f\in\mathcal{F}}f(X_i)\|_{\psi_2})^{2k}}{(2k)!}$$

$$= \exp\left(2(1+\beta)m\sigma^2\lambda^2\right) + \sqrt{\frac{(1+\beta)m\sigma^2\lambda^2}{2}}\left(\exp\left(2(1+\beta)m\sigma^2\lambda^2\right) - 1\right)$$

$$+ \frac{C\lambda\left\|\max_i\sup_{f\in\mathcal{F}}f(X_i)\right\|_{\psi_2}}{2}\left(\exp\left(\left(C\lambda\left\|\max_i\sup_{f\in\mathcal{F}}f(X_i)\right\|_{\psi_2}\right)^2\right) - 1\right)$$

$$+ 3\exp\left(\left(C\lambda\left\|\max_i\sup_{f\in\mathcal{F}}f(X_i)\right\|_{\psi_2}\right)^2\right)\right)$$

$$\leq \exp\left(4(1+\beta)m\sigma^2\lambda^2\right) + 3\exp\left(2\left(C\lambda\left\|\max_i\sup_{f\in\mathcal{F}}f(X_i)\right\|_{\psi_2}\right)^2\right),$$

where the first inequality follows from the inequality that $2(k!)^2 \leq (2k)!$.

The proof is complete.

$\square$

*Proof of Lemma 6.* According to Theorem 10, we have

$$\mathbb{P}(|Q - (1+\eta)\mathbb{E}Q| \geq t) \leq 2\exp\left(-\frac{t^2}{2(1+\beta)m\sigma^2}\right) + 6\exp\left(-\frac{t}{C\|\max_i\sup_{f\in\mathcal{F}}|f(X_i)|\|_{\psi_1}}\right).$$

Similarly, let the random variable $Y = Q - (1+\eta)\mathbb{E}Q$ we have that for any $k \geq 1$,

$$\mathbb{E}[|Y|^k]$$

$$= \int_0^{\infty}\mathbb{P}\left(|Y|^k > t\right)dt$$

$$= \int_0^{\infty}\mathbb{P}\left(|Y| > t^{1/k}\right)dt$$

$$\leq \int_0^{\infty}2\exp\left(-\frac{t^{2/k}}{2(1+\beta)m\sigma^2}\right)dt + \int_0^{\infty}6\exp\left(-\frac{t^{1/k}}{C\|\max_i\sup_{f\in\mathcal{F}}f(X_i)\|_{\psi_1}}\right)dt$$

$$= \left(2(1+\beta)m\sigma^2\right)^{k/2}k\int_0^{\infty}e^{-u}u^{k/2-1}du + 6k\left(C\left\|\max_i\sup_{f\in\mathcal{F}}f(X_i)\right\|_{\psi_1}\right)^k\int_0^{\infty}e^{-v}v^{k-1}dv$$

$$\leq \left(2(1+\beta)m\sigma^2\right)^{k/2}k\Gamma(k/2) + 6k\left(C\left\|\max_i\sup_{f\in\mathcal{F}}f(X_i)\right\|_{\psi_1}\right)^k\Gamma(k),$$

where we denote $u = \frac{t^{2/k}}{2(1+\beta)m\sigma^2}$ and $v = \frac{t^{1/k}}{C\|\max_i\sup_{f\in\mathcal{F}}f(X_i)\|_{\psi_1}}$ in the third equality.

Next, we use the Taylor expansion of the exponential function as follows. For $0 \leq \lambda \leq \frac{1}{2C\|\max_i \sup_{f\in\mathcal{F}} f(X_i)\|_{\psi_1}}$, we have

$$\mathbb{E}\exp(\lambda Y)$$

$$=1+\sum_{k=2}^{\infty}\frac{\lambda^k \mathbb{E}[|Y|^k]}{k!}$$

$$\leq 1+\sum_{k=2}^{\infty}\frac{\left(2(1+\beta)m\sigma^2\lambda^2\right)^{k/2}k\Gamma(k/2)+6k(C\lambda\|\max_i\sup_{f\in\mathcal{F}}f(X_i)\|_{\psi_1})^k\Gamma(k)}{k!}$$

$$=1+\sum_{k=1}^{\infty}\frac{\left(2(1+\beta)m\sigma^2\lambda^2\right)^k 2k\Gamma(k)}{(2k)!}+\sum_{k=1}^{\infty}\frac{\left(2(1+\beta)m\sigma^2\lambda^2\right)^{k+1/2}(2k+1)\Gamma(k+1/2)}{(2k+1)!}$$

$$+\sum_{k=2}^{\infty}6\left(C\lambda\left\|\max_i\sup_{f\in\mathcal{F}}f(X_i)\right\|_{\psi_1}\right)^k$$

$$\leq 1+\left(2+\sqrt{2(1+\beta)m\sigma^2\lambda^2}\right)\sum_{k=1}^{\infty}\frac{\left(2(1+\beta)m\sigma^2\lambda^2\right)^k k!}{(2k)!}$$

$$+6\left(C\lambda\left\|\max_i\sup_{f\in\mathcal{F}}f(X_i)\right\|_{\psi_1}\right)^2\sum_{k=0}^{\infty}\left(C\lambda\left\|\max_i\sup_{f\in\mathcal{F}}f(X_i)\right\|_{\psi_1}\right)^k$$

$$\leq 1+\left(1+\sqrt{\frac{(1+\beta)m\sigma^2\lambda^2}{2}}\right)\sum_{k=1}^{\infty}\frac{\left(2(1+\beta)m\sigma^2\lambda^2\right)^k}{(2k)!}+12\left(C\lambda\left\|\max_i\sup_{f\in\mathcal{F}}f(X_i)\right\|_{\psi_1}\right)^2$$

$$=\exp\left(2(1+\beta)m\sigma^2\lambda^2\right)+\sqrt{\frac{(1+\beta)m\sigma^2\lambda^2}{2}}\left(\exp\left(2(1+\beta)m\sigma^2\lambda^2\right)-1\right)$$

$$+\exp\left(12\left(C\lambda\left\|\max_i\sup_{f\in\mathcal{F}}f(X_i)\right\|_{\psi_1}\right)^2\right)$$

$$\leq \exp\left(4(1+\beta)m\sigma^2\lambda^2\right)+\exp\left(12\left(C\lambda\left\|\max_i\sup_{f\in\mathcal{F}}f(X_i)\right\|_{\psi_1}\right)^2\right),$$

where the second equality satisfies because of commutative property of positive convergent series and the third inequality follows from the inequality that $2(k!)^2 \leq (2k)!$ and $0 \leq \lambda \leq \frac{1}{2C\|\max_i \sup_{f\in\mathcal{F}} f(X_i)\|_{\psi_1}}$.

The proof is complete.

$\square$

## B  PROOFS OF SECTION 3

*Proof of Theorem 1.* Let $\{U_1,\ldots,U_m\}$ and $\{W_1,\ldots,W_m\}$ be sampled uniformly from a finite set of $M$-dimensional vectors [2] $\{\boldsymbol{v}_1,\ldots,\boldsymbol{v}_N\}\subset\mathbb{R}^M$ with and without replacement respectively, where

---

[2] We assume that $\mathcal{F}$ is a countable class of functions and this can be translated to the uncountable classes. For instance, if the empirical process is separable, meaning that $\mathcal{F}$ contains a dense countable subset. We refer to page 314 of [3] or page 72 of [5]

$\boldsymbol{v}_j = (f_1(c_j), \ldots, f_M(c_j))^T$. According to Lemma 2 and Theorem 6, we get that for all $\lambda > 0$:

$$\mathbb{E}\left[e^{\lambda Q'_m}\right] = \mathbb{E}\left[\exp\left(\lambda \sup_{j=1,\ldots,M}\left(\sum_{i=1}^m W_i\right)\right)_j\right] \leq \mathbb{E}\left[\exp\left(\lambda \sup_{j=1,\ldots,M}\left(\sum_{i=1}^m u_i\right)\right)_j\right] = \mathbb{E}\left[e^{\lambda Q_m}\right],$$

(11)

where the lower index $j$ indicates the $j$-th coordinate of a vector. According to Lemma 5, the moment generalization function of $Q_m$ can be bounded, which we can derive the following inequalities

$$\mathbb{E}\left[e^{\lambda Q'_m}\right] \leq \mathbb{E}\left[e^{\lambda Q_m}\right] \leq \exp\left((1+\eta)\lambda\mathbb{E}[Q_m] + 4(1+\beta)m\sigma^2\lambda^2\right)$$

$$+ 3\exp\left((1+\eta)\lambda\mathbb{E}[Q_m] + 2\left(C\lambda\left\|\max_i \sup_{f\in\mathcal{F}} f(X_i)\right\|_{\psi_2}\right)^2\right)$$

or, equivalently,

$$\mathbb{E}\left[e^{\lambda(Q'_m - (1+\eta)\mathbb{E}[Q'_m])}\right]$$

$$\leq \exp\left((1+\eta)\lambda(\mathbb{E}[Q_m] - \mathbb{E}[Q'_m]) + 4(1+\beta)m\sigma^2\lambda^2\right)$$

$$+ 3\exp\left((1+\eta)\lambda(\mathbb{E}[Q_m] - \mathbb{E}[Q'_m]) + 2\left(C\lambda\left\|\max_i \sup_{f\in\mathcal{F}} f(X_i)\right\|_{\psi_2}\right)^2\right).$$

Using Chernoff's method, we can obtain that for all $\epsilon \geq 0$ and $\lambda > 0$:

$$\mathbb{P}\left\{Q'_m - (1+\eta)\mathbb{E}[Q'_m] \geq \epsilon\right\}$$

$$\leq \frac{\mathbb{E}\left[e^{\lambda(Q'_m - (1+\eta)\mathbb{E}[Q'_m])}\right]}{e^{\lambda\epsilon}}$$

$$\leq \frac{\exp\left((1+\eta)\lambda(\mathbb{E}[Q_m] - \mathbb{E}[Q'_m]) + 4(1+\beta)m\sigma^2\lambda^2\right)}{\exp(\lambda\epsilon)}$$

$$+ \frac{3\exp\left((1+\eta)\lambda(\mathbb{E}[Q_m] - \mathbb{E}[Q'_m]) + 2\left(C\lambda\left\|\max_i \sup_{f\in\mathcal{F}} f(X_i)\right\|_{\psi_2}\right)^2\right)}{\exp(\lambda\epsilon)}$$

$$\leq \frac{\exp\left((1+\eta)\lambda(\mathbb{E}[Q_m] - \mathbb{E}[Q'_m])\right)\left(\exp(4(1+\beta)m\sigma^2\lambda^2) + 3\exp\left(2\left(C\lambda\left\|\max_i \sup_{f\in\mathcal{F}} f(X_i)\right\|_{\psi_2}\right)^2\right)\right)}{\exp(\lambda\epsilon)}$$

$$\leq 6\exp\left(\left((1+\eta)(\mathbb{E}[Q_m] - \mathbb{E}[Q'_m]) - \epsilon\right)\lambda + \left(4(1+\beta)m\sigma^2 + 2C^2\left\|\max_i \sup_{f\in\mathcal{F}} f(X_i)\right\|_{\psi_2}^2\right)\lambda^2\right),$$

(12)

where the first inequality applies Chernoff's method. The third hold under the following two terms $\exp\left(4(1+\beta)m\sigma^2\lambda^2\right) \geq 1$ and $\exp\left(2\left(C\lambda\left\|\max_i \sup_{f\in\mathcal{F}} f(X_i)\right\|_{\psi_2}\right)^2\right) \geq 1$. Using $a + b \leq 2ab$, $\forall a, b \geq 1$, we obtain the third inequality.

The term on the right-hand side of the last inequality achieves its minimum for

$$\lambda = \frac{\epsilon + (1+\eta)(\mathbb{E}[Q'_m] - \mathbb{E}[Q_m])}{8(1+\beta)m\sigma^2 + 4C^2\left\|\max_i \sup_{f\in\mathcal{F}} f(X_i)\right\|_{\psi_2}^2}.$$

(13)

Insert (13) into (12), when we have the technical condition $\epsilon \geq (1+\eta)(\mathbb{E}[Q_m] - \mathbb{E}[Q'_m])$ where $\mathbb{E}[Q_m] \geq \mathbb{E}[Q'_m]$ follows from Theorem 6 by exploiting the fact that the supremum is a convex function., we obtain the following inequality

$$\mathbb{P}\left\{Q'_m - (1+\eta)\mathbb{E}[Q_m] \geq \epsilon\right\} \leq 6\exp\left(-\frac{\epsilon^2}{16(1+\beta)m\sigma^2 + 8C^2\left\|\max_i \sup_{f\in\mathcal{F}} f(X_i)\right\|_{\psi_2}^2}\right).$$

The proof is complete.

$\square$

*Proof of Theorem 2.* The proof of Theorem 2 is similar with Theorem 1. Let two series of random variables $\{U_1, \ldots, U_m\}$ and $\{W_1, \ldots, W_m\}$ be sampled uniformly form a finite set of $M$-dimensional vectors $\{\boldsymbol{v}_1, \ldots, \boldsymbol{v}_N\} \subset \mathbb{R}^M$ with and without replacement respectively, where $\boldsymbol{v}_j = (f_1(c_j), \ldots, f_M(c_j))^T$. According to Lemma 2 and Theorem 6, we get that for all $\lambda \leq 0$:

$$\mathbb{E}\left[e^{\lambda Q'_m}\right] = \mathbb{E}\left[\exp\left(\lambda \sup_{j=1,\ldots,M}\left(\sum_{i=1}^m W_i\right)_j\right)\right] \leq \mathbb{E}\left[\exp\left(\lambda \sup_{j=1,\ldots,M}\left(\sum_{i=1}^m u_i\right)_j\right)\right] = \mathbb{E}\left[e^{\lambda Q_m}\right],$$
(14)

where the lower index $j$ indicates the $j$-th coordinate of a vector. According to Lemma 6, the moment generalization function of $Q_m$ can be bounded, which we can derive the following inequalities

$$\mathbb{E}\left[e^{\lambda Q'_m}\right] \leq \mathbb{E}\left[e^{\lambda Q_m}\right] \leq \exp\left((1+\eta)\lambda\mathbb{E}[Q_m] + 4(1+\beta)m\sigma^2\lambda^2\right)$$

$$+ \exp\left((1+\eta)\lambda\mathbb{E}[Q_m] + 12\left(C\lambda\left\|\max_i \sup_{f \in \mathcal{F}} f(X_i)\right\|_{\psi_1}\right)^2\right)$$

or, equivalently,

$$\mathbb{E}\left[e^{\lambda(Q'_m - (1+\eta)\mathbb{E}[Q'_m])}\right]$$

$$\leq \exp\left((1+\eta)\lambda(\mathbb{E}[Q_m] - \mathbb{E}[Q'_m]) + 4(1+\beta)m\sigma^2\lambda^2\right)$$

$$+ \exp\left((1+\eta)\lambda(\mathbb{E}[Q_m] - \mathbb{E}[Q'_m]) + 12\left(C\lambda\left\|\max_i \sup_{f \in \mathcal{F}} f(X_i)\right\|_{\psi_1}\right)^2\right).$$

Using Chernoff's method, we can obtain that for all $\epsilon \geq 0$ and $0 \leq \lambda \leq \frac{1}{2C\left\|\max_i \sup_{f \in \mathcal{F}} f(X_i)\right\|_{\psi_1}}$:

$$\mathbb{P}\left\{Q'_m - (1+\eta)\mathbb{E}[Q'_m] \geq \epsilon\right\}$$

$$\leq \frac{\mathbb{E}\left[e^{\lambda(Q'_m - (1+\eta)\mathbb{E}[Q'_m])}\right]}{e^{\lambda\epsilon}}$$

$$\leq \frac{\exp\left((1+\eta)\lambda(\mathbb{E}[Q_m] - \mathbb{E}[Q'_m]) + 4(1+\beta)m\sigma^2\lambda^2\right)}{\exp(\lambda\epsilon)}$$

$$+ \frac{\exp\left((1+\eta)\lambda(\mathbb{E}[Q_m] - \mathbb{E}[Q'_m]) + 12\left(C\lambda\left\|\max_i \sup_{f \in \mathcal{F}} f(X_i)\right\|_{\psi_1}\right)^2\right)}{\exp(\lambda\epsilon)}$$

$$\leq \frac{\exp\left((1+\eta)\lambda(\mathbb{E}[Q_m] - \mathbb{E}[Q'_m])\right)\left(\exp(4(1+\beta)m\sigma^2\lambda^2) + \exp\left(12\left(C\lambda\left\|\max_i \sup_{f \in \mathcal{F}} f(X_i)\right\|_{\psi_2}\right)^2\right)\right)}{\exp(\lambda\epsilon)}$$

$$\leq 2\exp\left(((1+\eta)(\mathbb{E}[Q_m] - \mathbb{E}[Q'_m]) - \epsilon)\lambda + \left(4(1+\beta)m\sigma^2 + 12C^2\left\|\max_i \sup_{f \in \mathcal{F}} f(X_i)\right\|_{\psi_1}^2\right)\lambda^2\right),$$
(15)

where the first inequality applies Chernoff's method and the third hold under the following two terms $\exp\left(4(1+\beta)m\sigma^2\lambda^2\right) \geq 1$ and $\exp\left(2\left(C\lambda\left\|\max_i \sup_{f \in \mathcal{F}} f(X_i)\right\|_{\psi_2}\right)^2\right) \geq 1$. Using $a + b \leq 2ab, \forall a, b \geq 1$, we obtain the third inequality.

The term on the right-hand side of the last inequality achieves its minimum for

$$\lambda = \frac{\epsilon + (1+\eta)(\mathbb{E}[Q'_m] - \mathbb{E}[Q_m])}{8(1+\beta)m\sigma^2 + 24C^2\left\|\max_i \sup_{f \in \mathcal{F}} f(X_i)\right\|_{\psi_1}^2}.$$
(16)

Insert (16) into (15), when we have the technical condition $(1 + \eta)(\mathbb{E}[Q_m] - \mathbb{E}[Q'_m]) \le \epsilon \le 12C \left\| \max_i \sup_{f \in \mathcal{F}} f(X_i) \right\|_{\psi_1}$, we obtain the following inequality

$$\mathbb{P}\{Q'_m - (1 + \eta)\mathbb{E}[Q_m] \ge \epsilon\} \le 2 \exp\left( - \frac{\epsilon^2}{16(1 + \beta)m\sigma^2 + 48C^2 \left\| \max_i \sup_{f \in \mathcal{F}} f(X_i) \right\|_{\psi_1}^2} \right).$$

The proof is complete.

$\square$

## C  PROOFS OF SECTION 4

### C.1  PROOFS OF SUBSECTION 4.1

From now on it will be convenient to introduce the following operators, mapping functions $f$ defined on $\boldsymbol{X}_N$ to $\mathbb{R}$:

$$Ef = \frac{1}{N} \sum_{i=1}^{N} f(\mathbf{x}_i), \mathbf{x}_i \in \boldsymbol{X}_N, \quad E_m f = \frac{1}{N} \sum_{\mathbf{x}_j=1}^{m} f(\mathbf{x}_j), \mathbf{x}_j \in \boldsymbol{X}_m.$$

Assume that there is a function $\mathbf{w}_N^* \in \mathcal{W}$ satisfying $R_N(\mathbf{w}_N^*) = \inf_{\mathbf{w} \in \mathcal{W}} R_N(\mathbf{w})$. Define the excess loss class $\mathcal{F}^* = \{f : f(\mathbf{x}) = \ell(\mathbf{w}; \mathbf{x}) - \ell(\mathbf{w}_N^*; \mathbf{x}), \mathbf{w} \in \mathcal{W}\}$.

Let $\{\xi_1, \ldots, \xi_n\}$ be random variables sampled with replacement from $\boldsymbol{X}_N$. The mapping functions $f$ defined on $\boldsymbol{X}_N$ to $\mathbb{R}$. Denote

$$E_{r,m}f = \mathbb{E}\left[ \sup_{f \in \mathcal{F}^*: Ef^2 \le r} \left( Ef - \frac{1}{m}\sum_{i=1}^{m} f(\xi_i) \right) \right]. \tag{17}$$

Then we have

$$\begin{aligned}
E_{r,m}f =& \mathbb{E}\left[ \sup_{f \in \mathcal{F}^*: Ef^2 \le r} \left( Ef - \frac{1}{m}\sum_{i=1}^{m} f(\xi_i) \right) \right] \\
\le& 2\mathbb{E}_{\xi \sim \boldsymbol{X}_N, v}\left[ \sup_{f \in \mathcal{F}^*: Ef^2 \le r} v_i \left( Ef - \frac{1}{m}\sum_{i=1}^{m} f(\xi_i) \right) \right] \\
\le& 2\mathbb{E}_v\left[ \sup_{f \in \mathcal{F}^*: Ef^2 \le r} \sum_{i=1}^{m} v_i Ef \right] + 2\mathbb{E}_{\xi \sim \boldsymbol{X}_N, v}\left[ \sup_{f \in \mathcal{F}^*: Ef^2 \le r} \frac{1}{m}\sum_{i=1}^{m} v_i f(\xi_i) \right] \\
=& 2\mathfrak{R}_N\{f \in \mathcal{F}^* : Ef^2 \le r\}.
\end{aligned}$$

where the first inequality holds using symmetrization inequality (see Lemma 11.4 [3])

**Lemma 7** (Peeling Lemma for sub-Gaussian). *Assume that there is a constant $B > 0$ such that for every $f \in \mathcal{F}^*$ we have $Ef^2 \le B \cdot Ef$. Suppose Assumptions 1 hold and the objective function $\ell(\cdot; \cdot)$ is sub-Gaussian.. Assume there is a sub-root function $\psi_m(r)$ such that*

$$2B\mathfrak{R}_N\{f \in \mathcal{F}^* : Ef^2 \le r\} \le \psi_m(r),$$

*where $E_{r,m}$ was defined in (17). Let $r_m^*$ be a fixed point of $\psi_m(r)$.*

*Fix some $\lambda > 1$. For $w(r, f) = \min\{r\lambda^k : k \in \mathbb{N}, r\lambda^k \ge Ef^2\}$, define the following rescaled version of excess loss class:*

$$\mathcal{G}_r = \left\{ \frac{r}{w(r, f)} f : f \in \mathcal{F}^* \right\}.$$

*Then for any $r > r_m^*$ and $t > 0$, with probability at least $1 - \delta$, we have*

$$\sup_{g \in \mathcal{G}_r} Eg - E_m g \leq \frac{(1+\eta)\sqrt{rr_m^*}}{B} \left( 1 + \frac{1}{K_2 \sqrt{\log \frac{2}{\delta}}} \right)$$

$$+ 4\sqrt{(1+\beta)\left(\frac{N}{m^2}\right) r \log \frac{12}{\delta}} + 4\sqrt{\frac{2C^2 K \log N}{m^2} \log \frac{12}{\delta}},$$

*where $K, K_2, \eta, \beta$ are some positive constants. $C$ is positive constants depending on $\eta, \beta$.*

*Proof of Lemma 7.* We use traditional peeling technologies presented in the proof of the first part of Theorem 3.3 of [2], but using Theorem 1 in place of Talagrand's inequality.

Firstly, for any $f \in \mathcal{F}^*$, we have

$$\mathbb{V}[f(\mathbf{x})] = Ef^2 - (Ef)^2 \leq Ef^2. \tag{18}$$

Let us fix some $\lambda > 1$ and $r > 0$ and introduce the following rescaled version of excess loss class:

$$\mathcal{G}_r = \left\{ \frac{r}{w(r, f)} f : f \in \mathcal{F}^* \right\},$$

where $w(r, f) = \min\{r\lambda^k : k \in \mathbb{N}, r\lambda^k \geq Ef^2\}$.

Let us consider functions $f \in \mathcal{F}^*$ such that $Ef^2 < r$, meaning $w(r, f) = r$. The functions $g \in \mathcal{G}_r$ corresponding to those functions satisfy $g = f$ and thus $\mathbb{V}[g(\mathbf{x})] = \mathbb{V}[f(\mathbf{x})] \leq Ef^2 \leq r$. Otherwise, if $Ef^2 > r$, then $w(r, f) = \lambda^k r$, and thus the functions $g \in \mathcal{G}_r$ corresponding to them satisfy $g = \frac{f}{\lambda^k}$ and $Ef^2 \in (r\lambda^{k-1}, r\lambda^k]$. Thus we have $\mathbb{V}[g(\mathbf{x})] = \frac{\mathbb{V}[f(\mathbf{x})]}{\lambda^{2k}} \leq \frac{Ef^2}{\lambda^{2k}} \leq r$. We conclude that, for any $g \in \mathcal{G}_r$, it holds $\mathbb{V}[g(X)] \leq r$.

Next we need to upper bound the following quantity:

$$V_r = \sup_{g \in \mathcal{G}_r} Eg - E_m g.$$

Note that any $f \in \mathcal{F}^*$, $f(\mathbf{x})$ is sub-Gaussian, thus for all $g \in \mathcal{G}_r$, $g(\mathbf{x})$ is sub-Gaussian. Notice that

$$\frac{1}{2}(Eg - E_m g) = \frac{1}{m} \sum_{\mathbf{x} \in \boldsymbol{X}_m} \frac{Eg - g(\mathbf{x})}{2}.$$

Note that $(Eg - g(\mathbf{x}))/2$ is also sub-Gaussian and $\mathbb{E}[Eg - g(\mathbf{x})] = 0$. Since $Eg$ is not random, using (18), for all $g \in \mathcal{G}_r$ we also have

$$\mathbb{V}\left[ \frac{Eg - g(\mathbf{x})}{2} \right] = \frac{\mathbb{V}[g(\mathbf{x})]}{4} \leq \frac{r}{4},$$

Besides, we need to bound $\left\| \max_{\mathbf{x}} \sup_{g \in \mathcal{G}_r} \frac{Eg - g(\mathbf{x})}{2} \right\|_{\psi_2}^2$.

$$\left\| \max_{\mathbf{x}} \sup_{g \in \mathcal{G}_r} \frac{Eg - g(\mathbf{x})}{2} \right\|_{\psi_2}^2 = \frac{\left\| \max_{\mathbf{x}} \sup_f Ef - f(\mathbf{x}) \right\|_{\psi_2}^2}{4\lambda^{2k}}$$

$$\leq K^2 \max_{\mathbf{x}} \left\| \sup_f \ell(\mathbf{w}; \mathbf{x}) \right\|_{\psi_2}^2 \log N \leq K \log N,$$

where $K$ is a positive constant. The first inequality holds using Theorem [34] and the second inequality satisfies because $\ell(\cdot; \mathbf{x})$ is sub-Gaussian.

We can now apply either Theorem 1 for the following function class: $\{(Eg - g(\mathbf{x}))/2, g \in \mathcal{G}_r\}$. Here we present the proof based on Theorem 1. Applying it we get that for all $\delta \in (0, 1)$, with probability at least $1 - \frac{\delta}{2}$, we have

$$
\frac{1}{2} \sup_{g \in \mathcal{G}_r} Eg - E_m g
$$

$$
\leq \frac{1 + \eta}{2} \mathbb{E}\left[\sup_{g \in \mathcal{G}_r} E_{r,m} g\right] + \sqrt{\left(16(1 + \beta)\left(\frac{N}{m^2}\right)\frac{1}{4}\sup_{g \in \mathcal{G}_r}\mathbb{V}[g(\mathbf{x})] + \frac{8C^2 K \log N}{m^2}\right)\log\frac{12}{\delta}}
$$

$$
\leq \frac{1 + \eta}{2} \mathbb{E}\left[\sup_{g \in \mathcal{G}_r} E_{r,m} g\right] + \sqrt{\left(4(1 + \beta)\left(\frac{N}{m^2}\right)r + \frac{8C^2 K \log N}{m^2}\right)\log\frac{12}{\delta}}
$$

$$
\leq \frac{1 + \eta}{2} \mathbb{E}\left[\sup_{g \in \mathcal{G}_r} E_{r,m} g\right] + 2\sqrt{(1 + \beta)\left(\frac{N}{m^2}\right)r \log\frac{12}{\delta}} + 2\sqrt{\frac{2C^2 K \log N}{m^2}\log\frac{12}{\delta}},
$$

where the last inequality holds because $\sqrt{a + b} \leq \sqrt{a} + \sqrt{b}$ for any $a \geq 0$ and $b \geq 0$.

Rewriting above inequality we have

$$
Vr \leq (1 + \eta)\mathbb{E}\left[\sup_{g \in \mathcal{G}_r} E_{r,m} g\right] + 4\sqrt{(1 + \beta)\left(\frac{N}{m^2}\right)r \log\frac{12}{\delta}} + 4\sqrt{\frac{2C^2 K \log N}{m^2}\log\frac{12}{\delta}}. \tag{19}
$$

Now we set $\mathcal{F}^*(x, y) = \{f \in \mathcal{F}^* : x \leq Ef^2 \leq y\}$, Note that $Ef$ is sub-Gaussian, for $f \in \mathcal{F}^*$, for any $\delta \in (0, 1)$ with probability at least $1 - \frac{\delta}{2}$, we have $\mathbb{V}[f(\mathbf{x})] \leq Ef^2 \leq B \cdot Ef \leq BK_2\sqrt{\log 2/\delta}$. Define $k$ to be the smallest integer such that $r\lambda^{k+1} \leq BK_2\sqrt{\log 2/\delta}$. Notice that, for any sets $A$ and $B$, we have:

$$
\mathbb{E}\left[\sup_{g \in A \cup B} E_{r,m} g\right] \leq \mathbb{E}\left[\sup_{g \in A} E_{r,m} g\right] + \mathbb{E}\left[\sup_{g \in B} E_{r,m} g\right]
$$

Since supremum is a convex function, we can use Jensen's inequality to show that each of the terms is positive. Then for any $\delta \in (0, 1)$, with probability at least $1 - \frac{\delta}{2}$, we have:

$$
\mathbb{E}\left[\sup_{g \in \mathcal{G}_r} E_{r,m} g\right]
$$

$$
\leq \mathbb{E}\left[\sup_{f \in \mathcal{F}^*(0,r)} E_{r,m} f\right] + \mathbb{E}\left[\sup_{f \in \mathcal{F}^*(r, 2BK_2\sqrt{2\log 2/\delta})} \frac{r}{w(r, f)} E_{r,m} f\right]
$$

$$
\leq \mathbb{E}\left[\sup_{f \in \mathcal{F}^*(0,r)} E_{r,m} f\right] + \sum_{i=0}^{k} \mathbb{E}\left[\sup_{f \in \mathcal{F}^*(r\lambda^i, r\lambda^{i+1})} \frac{r}{w(r, f)} E_{r,m} f\right]
$$

$$
\leq \mathbb{E}\left[\sup_{f \in \mathcal{F}^*(0,r)} E_{r,m} f\right] + \sum_{i=0}^{k} \lambda^{-i} \mathbb{E}\left[\sup_{f \in \mathcal{F}^*(r\lambda^i, r\lambda^{i+1})} E_{r,m} f\right]
$$

$$
\leq 2\mathfrak{R}_N\{f \in \mathcal{F}^* : Ef^2 \leq r\} + 2\sum_{i=0}^{k} \lambda^{-i} \mathfrak{R}_N\{f \in \mathcal{F}^* : r\lambda^i \leq Ef^2 \leq r\lambda^{i+1}\}
$$

$$
\leq \frac{\psi_m(r)}{B} + \frac{1}{BK_2\sqrt{\log\frac{2}{\delta}}} \sum_{i=0}^{k} \lambda^{-i} \psi_m(r\lambda^{i+1}),
$$

where the last inequality satisfies because $Ef$ is sub-Gaussian. Next, since $\psi_m$ is sub-root, for any $\beta \geq 1$, we have $\psi_m(\beta r) \leq \sqrt{\beta}\psi_m(r)$. Thus

$$
\mathbb{E}[V_r] \leq \sqrt{\beta} \leq \frac{\psi_m(r)}{B}\left(1 + \frac{\sqrt{\lambda}}{K_2\sqrt{\log\frac{2}{\delta}}} \sum_{i=0}^{k} \lambda^{-i/2}\right).
$$

Taking $\lambda = 4$, the right hand side is upper bounded by $\frac{\psi_m(r)}{B}\left(1 + \frac{1}{K_2\sqrt{\log\frac{2}{\delta}}}\right)$. Finally we note that for $r \geq r_m^*$, then for all $r \geq r_m^*$, it holds $\psi_m(r) \leq \sqrt{r/r_m^*}\psi_m(r_m^*) = \sqrt{rr_m^*}$. Thus, for any $\delta \in (0,1)$, with probability at least $1 - \frac{\delta}{2}$

$$\mathbb{E}\left[\sup_{g\in\mathcal{G}_r} E_{r,m}g\right] \leq \frac{\sqrt{rr_m^*}}{B}\left(1 + \frac{1}{K_2\sqrt{1\log\frac{2}{\delta}}}\right). \tag{20}$$

Combining (20) and (19), according to the union bound, for any $\delta \in (0,1)$, with probability at least $1 - \delta$, we have

$$\sup_{g\in\mathcal{G}_r} Eg - E_mg \leq \frac{(1+\eta)\sqrt{rr_m^*}}{B}\left(1 + \frac{1}{K_2\sqrt{\log\frac{2}{\delta}}}\right)$$
$$+ 4\sqrt{(1+\beta)\left(\frac{N}{m^2}\right)r\log\frac{12}{\delta}} + 4\sqrt{\frac{2C^2K\log N}{m^2}\log\frac{12}{\delta}},$$

where $K, K_2, \eta, \beta$ are some positive constants. $C$ is positive constants depending on $\eta, \beta$.

The proof is complete.

$\square$

**Lemma 8.** *Under the assumptions of Theorem 3, for any $\delta \in (0,1)$, with probability at least $1 - \delta$, we have*

$$R_N(\hat{\mathbf{w}}_m) - R_N(\mathbf{w}_N^*) \leq \frac{c_1 r_m^*}{B\log\frac{2}{\delta}} + \frac{c_2 N\log\frac{12}{\delta}}{m^2} + \frac{c_3\sqrt{\log N\log\frac{12}{\delta}}}{m},$$

*where $c_1, c_2$ and $c_3$ are some positive constants.*

*Proof of Lemma 8.* According to Lemma 7, we have the following results that, for any $r > r_m^*$, $\delta \in (0,1)$ and $\lambda > 1$, with probability at least $1 - \delta$, we have

$$\sup_{g\in\mathcal{G}_r} Eg - E_mg \leq \frac{(1+\eta)\sqrt{rr_m^*}}{B}\left(1 + \frac{1}{K_2\sqrt{\log\frac{2}{\delta}}}\right)$$
$$+ 4\sqrt{(1+\beta)\left(\frac{N}{m^2}\right)r\log\frac{12}{\delta}} + 4\sqrt{\frac{2C^2K\log N}{m^2}\log\frac{12}{\delta}}, \tag{21}$$

where $\mathcal{G}_r$ is the rescaled excess loss class:

$$\mathcal{G}_r = \left(\frac{r}{w(r,f)}f : f \in \mathcal{F}^*\right),$$

and $w(r,f) = \min\{r\lambda^k : k \in \mathbb{N}, r\lambda^k \geq Ef^2\}$. Now we want to choose $r_0 > r_m^*$ in such a way that the upper bound of (21) becomes of a form $\frac{r_0}{\lambda BK'}$, we achieve this by setting:

$$r_0 = K'^2\lambda^2\left((1+\eta)\sqrt{r_m^*}\left(1 + \frac{1}{K_2\sqrt{\log\frac{2}{\delta}}}\right) + 4B\sqrt{(1+\beta)\left(\frac{N}{m^2}\right)\log\frac{12}{\delta}}\right)^2 > r_m^*.$$

Inserting $r = r_0$ into (21), we have

$$\sup_{g\in\mathcal{G}_{r_0}} Eg - E_mg \leq \frac{r_0}{\lambda BK'} + 4\sqrt{\frac{2C^2K\log N}{m^2}\log\frac{12}{\delta}}. \tag{22}$$

Further, using inequality $(u+v)^2 \leq 2(u^2 + v^2)$, we have

$$r_0 \leq 2(1+\eta)^2 \left(1 + \frac{1}{K_2\sqrt{\log\frac{2}{\delta}}}\right)^2 K'^2\lambda^2 r_m^* + 32(1+\beta)\left(\frac{N}{m^2}\right)K'^2\lambda^2 B^2 \log\frac{12}{\delta}. \quad (23)$$

Recall that for any $r > 0$ and all $g \in \mathcal{G}_r$, the following holds with probability 1

$$Eg - E_m g \leq \sup_{g \in \mathcal{G}_r} Eg - E_m g.$$

Using the definition of $\mathcal{G}_r$, for all $f \in \mathcal{F}^*$, with probability 1, we have the following inequality

$$E\left(\frac{r}{w(r,f)}f\right) - E_m\left(\frac{r}{w(r,f)}f\right) \leq \sup_{g \in \mathcal{G}_r} Eg - E_m g,$$

or, rewriting

$$Ef - E_m f \leq \frac{w(r,f)}{r}\sup_{g \in \mathcal{G}_r} Eg - E_m g.$$

Next we setting $r = r_0$ and using (22), for any $\delta \in (0,1)$, with probability at least $1 - \delta$, we have

$$\forall f \in \mathcal{F}^*, \forall K > 1: \quad Ef - E_m f \leq \frac{w(r_0, f)}{r_0}\left(\frac{r_0}{\lambda K' B} + 4\sqrt{\frac{2C^2 K \log N}{m^2}\log\frac{12}{\delta}}\right).$$

Next, according to $Ef^2 \leq B \cdot Ef$, if for $f \in \mathcal{F}^*$, $Ef^2 \leq r_0$, we have $w(r_0, f) = r_0$ and using (23), we have

$$Ef - E_m f \leq \frac{w(r_0, f)}{r_0}\left(\frac{r_0}{\lambda K' B} + 4\sqrt{\frac{2C^2 K \log N}{m^2}\log\frac{12}{\delta}}\right)$$

$$\leq \frac{2(1+\eta)^2 K'\lambda r_m^*}{B}\left(1 + \frac{1}{K_2\sqrt{\log\frac{2}{\delta}}}\right)^2 + 32(1+\beta)\left(\frac{N}{m^2}\right)K'\lambda B\log\frac{12}{\delta} + 4\sqrt{\frac{2C^2 K \log N}{m^2}\log\frac{12}{\delta}}.$$

Rewriting,

$$Ef \leq E_m f + \frac{2(1+\eta)^2 K'\lambda r_m^*}{B}\left(1 + \frac{1}{K_2\sqrt{\log\frac{2}{\delta}}}\right)^2$$

$$+ 32(1+\beta)\left(\frac{N}{m^2}\right)K'\lambda B\log\frac{12}{\delta} + 4\frac{CK_2\sqrt{2\log\frac{12}{\delta}}}{m}. \quad (24)$$

On the other hand, if $Ef^2 > r_0$, then $w(r_0, f) = \lambda^i r_0$ for certain value of $i > 0$ and also $Ef^2 \in (r_0\lambda^{i-1}, r_0\lambda^i]$. Then we have

$$Ef - E_m f$$

$$\leq \frac{w(r_0, f)}{r_0}\left(\frac{r_0}{\lambda K' B} + 4\sqrt{\frac{2C^2 K \log N}{m^2}\log\frac{12}{\delta}}\right)$$

$$\leq \frac{\lambda^{i-1}r_0}{K'B} + \frac{4\lambda^{i-1}\sqrt{2C^2 K \log N \log\frac{12}{\delta}}}{m}$$

$$\leq \frac{Ef^2}{K'B} + \frac{4\lambda^{i-1}\sqrt{2C^2 K \log N \log\frac{12}{\delta}}}{m}$$

$$\leq \frac{Ef}{K'} + \frac{4\lambda^{i-1}\sqrt{2C^2 K \log N \log\frac{12}{\delta}}}{m}.$$

Thus, we have

$$Ef \leq \frac{K'}{K'-1} E_m f + \frac{4K'\lambda^{i-1}\sqrt{2C^2 K \log N \log \frac{12}{\delta}}}{(K'-1)m}. \tag{25}$$

Combing (24) and (25), for any $\delta \in (0,1)$, with probability at least $1 - \delta$, we have

$$\forall f \in \mathcal{F}^*, \forall K > 1: \quad Ef \leq \inf_{K'>1} \frac{K'}{K'-1} E_m f + \frac{2(1+\eta)^2 K' \lambda r_m^*}{B} \left(1 + \frac{1}{K_2 \sqrt{\log \frac{2}{\delta}}}\right)^2 \tag{26}$$

$$+32(1+\beta)\left(\frac{N}{m^2}\right) K'\lambda B \log \frac{12}{\delta} + 4\frac{CK_2\sqrt{2\log\frac{12}{\delta}}}{m} + \frac{4K'\lambda^{i-1}\sqrt{2C^2 K \log N \log \frac{12}{\delta}}}{(K-1)m}.$$

Finally we recall that the definition of $\mathcal{F}^*$ and put $\hat{f}_m(\cdot) = \ell(\hat{\mathbf{w}}_m; \cdot) - \ell(\mathbf{w}_N^*; \cdot)$. Notice that

$$E_m \hat{f}_m = E_m \ell(\hat{\mathbf{w}}_m) - E_m \ell(\mathbf{w}_N^*) = \hat{R}_m(\hat{\mathbf{w}}_m) - \hat{R}_m(\mathbf{w}_N^*) \leq 0,$$

and

$$E\hat{f}_m = R_N(\hat{\mathbf{w}}_m) - R_N(\mathbf{w}_N^*),$$

thus, we have

$$R_N(\hat{\mathbf{w}}_m) - R_N(\mathbf{w}_N^*) \leq \frac{c_1 r_m^*}{B \log \frac{2}{\delta}} + \frac{c_2 N \log \frac{12}{\delta}}{m^2} + \frac{c_3 \sqrt{\log N \log \frac{12}{\delta}}}{m},$$

where $c_1, c_2$ and $c_3$ are some positive constants.

The proof is complete. $\qquad\square$

**Lemma 9.** *Under the assumptions of Theorem 3, for any $\delta \in (0,1)$, with probability at least $1 - \delta$, we have*

$$R_u(\hat{\mathbf{w}}_m) - R_u(\mathbf{w}_u^*) \leq \frac{N}{u} \left( \frac{c_1 r_m^*}{B \log \frac{2}{\delta}} + \frac{c_2 N \log \frac{12}{\delta}}{m^2} + \frac{c_3 \sqrt{\log N \log \frac{12}{\delta}}}{m} \right)$$

$$+ \frac{N}{m} \left( \frac{c_1 r_u^*}{B \log \frac{2}{\delta}} + \frac{c_2 N \log \frac{12}{\delta}}{u^2} + \frac{c_3 \sqrt{\log N \log \frac{12}{\delta}}}{u} \right),$$

*where $c_1, c_2$ and $c_3$ are some positive constants.*

*Proof of Lemma 9.* Note that since $\mathbf{w}_u^*$ is also an empirical risk minimizer computed on the test set., the results of Lemma 8 also hold for $\mathbf{w}_u^*$ with every $m$ in the statement replaced by $u$. Also note that the following holds almost surely:

$$\begin{aligned}
0 &\leq R_N(\hat{\mathbf{w}}_m) - R_N(\mathbf{w}_N^*) \\
&= R_N(\hat{\mathbf{w}}_m) - R_N(\mathbf{w}_N^*) - \hat{R}_m(\hat{\mathbf{w}}_m) + \hat{R}_m(\mathbf{w}_N^*) + \hat{R}_m(\hat{\mathbf{w}}_m) - \hat{R}_m(\mathbf{w}_N^*) \\
&\leq R_N(\hat{\mathbf{w}}_m) - R_N(\mathbf{w}_N^*) - \hat{R}_m(\hat{\mathbf{w}}_m) + \hat{R}_m(\mathbf{w}_N^*) \\
&= \frac{u}{n} \left( R_u(\hat{\mathbf{w}}_m) - R_u(\mathbf{w}_N^*) - \hat{R}_m(\hat{\mathbf{w}}_m) + \hat{R}_m(\mathbf{w}_N^*) \right)
\end{aligned} \tag{27}$$

and

$$\begin{aligned}
0 &\leq R_N(\hat{\mathbf{w}}_u) - R_N(\mathbf{w}_N^*) \\
&= R_N(\hat{\mathbf{w}}_u) - R_N(\mathbf{w}_N^*) - R_u(\hat{\mathbf{w}}_u) + R_u(\mathbf{w}_N^*) + R_u(\hat{\mathbf{w}}_u) - R_u(\mathbf{w}_N^*) \\
&\leq R_N(\hat{\mathbf{w}}_u) - R_N(\mathbf{w}_N^*) - R_u(\hat{\mathbf{w}}_u) + R_u(\mathbf{w}_N^*) \\
&= \frac{m}{n} \left( \hat{R}_m(\hat{\mathbf{w}}_u) - \hat{R}_m(\mathbf{w}_N^*) - R_u(\hat{\mathbf{w}}_u) + R_u(\mathbf{w}_N^*) \right),
\end{aligned} \tag{28}$$

where last equations in both cases use the equation $N \cdot R_N(\mathbf{w}) = m \cdot \hat{R}_m(\mathbf{w}) + u \cdot R_u(\mathbf{w})$.

Now we are going to use (26) obtained in the proof of Lemma 8. Using (27) and, subsequently, employing (26) for $f = \ell(\hat{\mathbf{w}}_m; \cdot) - \ell(\mathbf{w}_N^*; \cdot)$, where we subtract $E_m f$ for both sides of (26), for any $\delta \in (0, 1)$, with probability at least $1 - \frac{\delta}{2}$, we obtain:

$$0 \le R_u(\hat{\mathbf{w}}_m) - R_u(\mathbf{w}_N^*) - \hat{R}_m(\hat{\mathbf{w}}_m) + \hat{R}_m(\mathbf{w}_N^*)$$

$$\le \frac{N}{u}\left( \inf_{K'>1} \frac{K'}{K'-1}\hat{R}_m(\hat{\mathbf{w}}_m - \mathbf{w}_N^*) + \frac{2(1+\eta)^2 K'\lambda r_m^*}{B}\left(1 + \frac{1}{K_2\sqrt{\log \frac{4}{\delta}}}\right)^2 \right.$$

$$\left. + 32(1+\beta)\left(\frac{N}{m^2}\right)K'\lambda B \log \frac{24}{\delta} + 4\frac{CK_2\sqrt{2\log \frac{12}{\delta}}}{m} + \frac{4K'\lambda^{i-1}\sqrt{2C^2 K \log N \log \frac{24}{\delta}}}{(K-1)m} \right).$$

Similarly, the same argument can be used for $\mathbf{w}_u^*$, which gives that for any $\delta \in (0, 1)$, with probability at least $1 - \frac{\delta}{2}$, we obtain:

$$0 \le \hat{R}_m(\hat{\mathbf{w}}_u) - \hat{R}_m(\mathbf{w}_N^*) - R_u(\hat{\mathbf{w}}_u) + R_u(\mathbf{w}_N^*)$$

$$\le \frac{N}{m}\left( \inf_{K'>1} \frac{K'}{K'-1}R_u(\hat{\mathbf{w}}_u - \mathbf{w}_N^*) + \frac{2(1+\eta)^2 K'\lambda r_u^*}{B}\left(1 + \frac{1}{K_2\sqrt{\log \frac{4}{\delta}}}\right)^2 \right.$$

$$\left. + 32(1+\beta)\left(\frac{N}{u^2}\right)K'\lambda B \log \frac{24}{\delta} + 4\frac{CK_2\sqrt{2\log \frac{12}{\delta}}}{u} + \frac{4K'\lambda^{i-1}\sqrt{2C^2 K \log N \log \frac{24}{\delta}}}{(K-1)u} \right).$$

The union bound gives us that both inequalities hold simultaneously with probability at least $1 - \delta$, summing these two inequalities, we obtain

$$0 \le R_u(\hat{\mathbf{w}}_m) - R_u(\mathbf{w}_u^*) - \hat{R}_m(\hat{\mathbf{w}}_m) + \hat{R}_m(\mathbf{w}_u^*)$$

$$\le \frac{N}{u}\left( \frac{c_1 r_m^*}{B \log \frac{2}{\delta}} + \frac{c_2 N \log \frac{12}{\delta}}{m^2} + \frac{c_3\sqrt{\log N \log \frac{12}{\delta}}}{m} \right) + \frac{N}{m}\left( \frac{c_1 r_u^*}{B \log \frac{2}{\delta}} + \frac{c_2 N \log \frac{12}{\delta}}{u^2} + \frac{c_3\sqrt{\log N \log \frac{12}{\delta}}}{u} \right).$$

Using the fact the $\hat{\mathbf{w}}_m$ and $\mathbf{w}_u^*$ are the empirical risk minimizers on the training and test set, respectively, we finally get:

$$0 \le R_u(\hat{\mathbf{w}}_m) - R_u(\mathbf{w}_u^*)$$

$$\le \frac{N}{u}\left( \frac{c_1 r_m^*}{B \log \frac{2}{\delta}} + \frac{c_2 N \log \frac{12}{\delta}}{m^2} + \frac{c_3\sqrt{\log N \log \frac{12}{\delta}}}{m} \right) + \frac{N}{m}\left( \frac{c_1 r_u^*}{B \log \frac{2}{\delta}} + \frac{c_2 N \log \frac{12}{\delta}}{u^2} + \frac{c_3\sqrt{\log N \log \frac{12}{\delta}}}{u} \right),$$

where $c_1, c_2$ and $c_3$ are some positive constants.

The proof is completed.

$\square$

*Proof of Theorem 3.* Notice that $2B\mathfrak{R}_N\{f \in \mathcal{F}^* : Ef^2 \le r\} \le \psi_m(r)$, according to Assumption 1, we have $\log \mathcal{N}(\varepsilon, \mathcal{W}, L_2(\mathbb{P})) \le \mathcal{O}(\log(1/\varepsilon))$. Using Dudley's integral bound [35] to find $\psi_m$ and solving $r \le \mathcal{O}(B\psi_m(r))$, it is not hard to verify that

$$r^* \le \mathcal{O}\left( \frac{B^2 \log m}{m} \right).$$

Insert the solution $r^*$ into Lemma 9, for any $\delta \in (0, 1)$, with probability at least $1 - \delta$, we have

$$\varepsilon_u(\hat{\mathbf{w}}_m) = \mathcal{O}\left( \frac{N}{mu}\left( \log m + \log u + \frac{N \log \frac{1}{\delta}}{m} + \frac{N \log \frac{1}{\delta}}{u} + \sqrt{\log N \log \frac{1}{\delta}} \right) \right).$$

The proof is complete.

$\square$

The detailed proof of Theorem 4 is completely similar with Theorem 3, In consequence, we omit here and give the Lemmas for sub-exponential.

**Lemma 10** (Peeling Lemma for sub-exponential). *Assume that there is a constant $B > 0$ such that for every $f \in \mathcal{F}^*$ we have $Ef^2 \leq B \cdot Ef$. Suppose Assumptions 1 hold and the objective function $\ell(\cdot; \cdot)$ is sub-exponential. Assume there is a sub-root function $\psi_m(r)$ such that*

$$2B\mathfrak{R}_N\{f \in \mathcal{F}^* : Ef^2 \leq r\} \leq \psi_m(r),$$

*where $E_{r,m}$ was defined in (17). Let $r_m^*$ be a fixed point of $\psi_m(r)$.*

*Fix some $\lambda > 1$. For $w(r, f) = \min\{r\lambda^k : k \in \mathbb{N}, r\lambda^k \geq Ef^2\}$, define the following rescaled version of excess loss class:*

$$\mathcal{G}_r = \left\{ \frac{r}{w(r, f)} f : f \in \mathcal{F}^* \right\}.$$

*Then for any $r > r_m^*$ and $t > 0$, with probability at least $1 - \delta$, we have*

$$\sup_{g \in \mathcal{G}_r} Eg - E_m g \leq \frac{(1+\eta)\sqrt{rr_m^*}}{B} \left( 1 + \frac{1}{K_1 \log \frac{2}{\delta}} \right)$$

$$+ 4\sqrt{(1+\beta)\left(\frac{N}{m^2}\right) r \log \frac{12}{\delta}} + 8\sqrt{\frac{3C^2 K \log^2 N}{m^2} \log \frac{12}{\delta}},$$

*where $K, K_1, \eta, \beta$ are some positive constants. $C$ is positive constants depending on $\eta, \beta$.*

**Lemma 11.** *Under the assumptions of Theorem 4, for any $\delta \in (0, 1)$, with probability at least $1 - \delta$, we have*

$$R_N(\hat{\mathbf{w}}_m) - R_N(\mathbf{w}_N^*) \leq \frac{c_1 r_m^*}{B \log^2 \frac{2}{\delta}} + \frac{c_2 N \log \frac{12}{\delta}}{m^2} + \frac{c_3 \sqrt{\log^2 N \log \frac{12}{\delta}}}{m},$$

*where $c_1, c_2$ and $c_3$ are some positive constants.*

**Lemma 12.** *Under the assumptions of Theorem 4, for any $\delta \in (0, 1)$, with probability at least $1 - \delta$, we have*

$$R_u(\hat{\mathbf{w}}_m) - R_u(\mathbf{w}_u^*) \leq \frac{N}{u} \left( \frac{c_1 r_m^*}{B \log^2 \frac{2}{\delta}} + \frac{c_2 N \log \frac{12}{\delta}}{m^2} + \frac{c_3 \sqrt{\log^2 N \log \frac{12}{\delta}}}{m} \right)$$

$$+ \frac{N}{m} \left( \frac{c_1 r_u^*}{B \log^2 \frac{2}{\delta}} + \frac{c_2 N \log \frac{12}{\delta}}{u^2} + \frac{c_3 \sqrt{\log^2 N \log \frac{12}{\delta}}}{u} \right),$$

*where $c_1, c_2$ and $c_3$ are some positive constants.*

## C.2 SOME RESULTS FOR SUB-EXPONENTIAL FUNCTIONS IN SUBSECTION 4.2

**Theorem 11.** *Suppose Assumptions 2, 3, 4, and 5 hold. For any $\mathbf{w} \in \mathcal{W}$, let the loss function $\ell(\mathbf{w}; \cdot)$ be sub-exponential. Suppose that the step sizes $\{\eta_t\}$ satisfies $\eta_t = \frac{1}{t+t_0}$ such that $t_0 \geq \max\{(2P)^{1/\alpha}, 1\}$. For any $\delta \in (0, 1)$, with probability $1 - \delta$,*

*(a). If $\alpha \in (0, \frac{1}{2})$, we have*

$$R_u(\mathbf{w}^{(T+1)}) - \hat{R}_m(\mathbf{w}^{(T+1)}) = \mathcal{O}\left( L_{\mathcal{F}} \frac{\sqrt{N}}{u} \log^{\frac{1}{2}}(T) T^{\frac{1-2\alpha}{2}} \log\left(\frac{1}{\delta}\right) + \frac{N}{u} \sqrt{\frac{\log^3\left(\frac{1}{\delta}\right)}{m}} \right).$$

*(b). If $\alpha = \frac{1}{2}$, we have*

$$R_u(\mathbf{w}^{(T+1)}) - \hat{R}_m(\mathbf{w}^{(T+1)}) = \mathcal{O}\left( L_{\mathcal{F}} \frac{\sqrt{N}}{u} \log(T) \log\left(\frac{1}{\delta}\right) + \frac{N}{u} \sqrt{\frac{\log^3\left(\frac{1}{\delta}\right)}{m}} \right).$$

*(c). If $\alpha \in (\frac{1}{2}, 1]$, we have*

$$R_u(\mathbf{w}^{(T+1)}) - \hat{R}_m(\mathbf{w}^{(T+1)}) = \mathcal{O}\left(L_{\mathcal{F}}\frac{\sqrt{N}}{u}\log^{\frac{1}{2}}(T)\log\left(\frac{1}{\delta}\right) + \frac{N}{u}\sqrt{\frac{\log^3\left(\frac{1}{\delta}\right)}{m}}\right).$$

**Corollary 2.** *Suppose Assumptions 2, 3, 4, and 5 hold. For any $\mathbf{w} \in \mathcal{W}$, let the loss function $\ell(\mathbf{w}; \cdot)$ be sub-exponential. Suppose that the learning rate $\{\eta_t\}$ satisfies $\eta_t = \frac{2}{\mu(t+t_0)}$ such that $t_0 \geq \max\{\frac{2}{\mu}(2P)^{\frac{1}{\alpha}}, 1\}$. For any $\delta \in (0, 1)$, with probability $1 - \delta$,*

*(a). If $\alpha \in (0, \frac{1}{2})$, we have*

$$R_u(\mathbf{w}^{(T+1)}) - \hat{R}_m(\mathbf{w}^*) = \mathcal{O}\left(L_{\mathcal{F}}\frac{\sqrt{Nd}}{u}\log^{\frac{1}{2}}(T)T^{\frac{1}{2}-\alpha}\log\left(\frac{1}{\delta}\right) + \frac{N}{u}\sqrt{\frac{\log^3\left(\frac{1}{\delta}\right)}{m}} + \frac{1}{T^\alpha}\right),$$

*(b). If $\alpha = \frac{1}{2}$, we have*

$$R_u(\mathbf{w}^{(T+1)}) - \hat{R}_m(\mathbf{w}^*) = \mathcal{O}\left(L_{\mathcal{F}}\frac{\sqrt{Nd}}{u}\log(T)\log\left(\frac{1}{\delta}\right) + \frac{N}{u}\sqrt{\frac{\log^3\left(\frac{1}{\delta}\right)}{m}} + \frac{1}{T^\alpha}\right).$$

*(c). If $\alpha \in (\frac{1}{2}, 1)$, we have*

$$R_u(\mathbf{w}^{(T+1)}) - \hat{R}_m(\mathbf{w}^*) = \mathcal{O}\left(L_{\mathcal{F}}\frac{\sqrt{Nd}}{u}\log^{\frac{1}{2}}(T)\log(1/\delta) + \frac{N}{u}\sqrt{\frac{\log^3\left(\frac{1}{\delta}\right)}{m}} + \frac{1}{T^\alpha}\right).$$

*(d). If $\alpha = 1$, we have*

$$R_u(\mathbf{w}^{(T+1)}) - R_u(\mathbf{w}^*) = s\mathcal{O}\left(L_{\mathcal{F}}\frac{\sqrt{Nd}}{u}\log^{\frac{1}{2}}(T)\log(1/\delta) + \frac{N}{u}\sqrt{\frac{\log^3\left(\frac{1}{\delta}\right)}{m}} + \frac{\log(T)\log^3(1/\delta)}{T}\right).$$

### C.3 PROOFS OF SUBSECTION 4.2

*Proof of Theorem 5.* In order to obtain high-probability bounds with our new concentration inequalities, for the term $\sup_{f_{\mathbf{w}} \in \mathcal{F}_{\mathcal{W}}} \sum_{\mathbf{x} \in \mathbf{X}_m} f_{\mathbf{w}}(\mathbf{x}) = \sup_{\mathbf{w} \in \mathcal{W}} \sum_{\mathbf{x} \in \mathbf{X}_m} (R_N(\mathbf{w}) - \ell(\mathbf{w}; \mathbf{x})) = m \cdot \sup_{\mathbf{w} \in \mathcal{W}}(R_N(\mathbf{w}) - \hat{R}_m(\mathbf{w}))$, where we obtain a factor of $m$ in the equation because in Theorem 1 we considered unnormalized sums.

To use Theorem 1, we need to bound $\left\|\max_{\mathbf{x}} \sup_{f_{\mathbf{w}} \in \mathcal{F}_{\mathcal{W}}} f_{\mathbf{w}}(\mathbf{x})\right\|_{\psi_2}^2$, we have

$$\left\|\max_{\mathbf{x}} \sup_{f_{\mathbf{w}} \in \mathcal{F}_{\mathcal{W}}} f_{\mathbf{w}}(\mathbf{x})\right\|_{\psi_2}^2 \leq \left\|\max_{\mathbf{x}} \sup_{\mathbf{w} \in \mathcal{W}} \ell(\mathbf{w}; \mathbf{x})\right\|_{\psi_2}^2 \leq K^2 \max_{\mathbf{x}} \left\|\sup_{\mathbf{w} \in \mathcal{W}} \ell(\mathbf{w}; \mathbf{x})\right\|_{\psi_2}^2 \log N \leq K^2 K_2^2 \log N.$$

where $K$ and $K_2$ are two positive constants. The second inequality holds using Theorem 7 [34] and the last inequality satisfies because $\ell(\cdot; \mathbf{x})$ is sub-Gaussian, using property of the tail bound for sub-Gaussian distribution.

Then we turn to bound $\sigma_{\mathcal{W}}^2$. For any fixed $\mathbf{w} \in \mathcal{W}$ and any $\delta \in (0, 1)$, with at least probability $1 - \frac{\delta}{2}$, we have

$$\frac{1}{N}\sum_{\mathbf{x} \in \mathbf{Z}_N}(\ell(\mathbf{w}; \mathbf{x}) - R_N(\mathbf{w}))^2 = \frac{1}{N}\sum_{\mathbf{x} \in \mathbf{Z}_N}\ell(\mathbf{w}; \mathbf{x})^2 - R_N(\mathbf{w})^2 \leq \frac{1}{N}\sum_{\mathbf{x} \in \mathbf{Z}_N}\ell(\mathbf{w}; \mathbf{x})^2 \leq K\log\frac{2}{\delta},$$

where $K$ is a positive constant. the last inequality holds because $\ell(\cdot; \mathbf{x})$ is sub-Gaussian, then $\ell(\cdot; \mathbf{x})^2$ is sub-exponential, using property of the tail bound for sub-exponential distribution. Thus for any $\delta \in (0, 1)$, with at least probability $1 - \frac{\delta}{2}$, we have

$$\sigma_{\mathcal{W}}^2 = \sup_{\mathbf{w} \in \mathcal{W}}\left(\frac{1}{N}\sum_{\mathbf{x} \in \mathbf{Z}_N}(\ell(\mathbf{w}; \mathbf{x}) - R_N(\mathbf{w}))^2\right) \leq K\log\frac{2}{\delta} \tag{29}$$

According to Theorem 1, Let $Q_m = m \cdot (R_N(\mathbf{w}) - \hat{R}_m(\mathbf{w}))$, and combined with (29). For any $\delta \in (0, 1)$ with probability at least $1 - \delta$, we have

$$
\sup_{\mathbf{w} \in \mathcal{W}} (R_N(\mathbf{w}) - \hat{R}_m(\mathbf{w}))
$$

$$
\leq (1 + \eta) E_m + 2 \sqrt{\left( \frac{4(1 + \beta) K \log \frac{2}{\delta}}{m} + \frac{2 C^2 K^2 K_2^2 \log n}{m^2} \right) \log \frac{12}{\delta}}
$$

$$
\leq (1 + \eta) E_m + 4 \sqrt{\frac{(1 + \beta) K \log \frac{2}{\delta} \log \frac{12}{\delta}}{m}} + \frac{2 \sqrt{2 C^2 K^2 K_2^2 \log N \log \frac{12}{\delta}}}{m} \tag{30}
$$

$$
\leq (1 + \eta) E_m + 4 \sqrt{\frac{(1 + \beta) K}{m}} \log \frac{12}{\delta} + \frac{2 \sqrt{2 C^2 K^2 K_2^2 \log N \log \frac{12}{\delta}}}{m}.
$$

where the second inequality holds using $\sqrt{a + b} \leq \sqrt{a} + \sqrt{b}$.

Next, we need to bound the $E_m = \mathbb{E}\left[ \sup_{\mathbf{w} \in \mathcal{W}} \left( R_N(\mathbf{w}) - \frac{1}{m} \sum_{i=1}^{m} \ell(\mathbf{w}; \xi_i) \right) \right]$. We have

$$
E_m = \mathbb{E}\left[ \sup_{\mathbf{w} \in \mathcal{W}} \left( R_N(\mathbf{w}) - \frac{1}{m} \sum_{i=1}^{m} \ell(\mathbf{w}; \xi_i) \right) \right]
$$

$$
\leq 2 \mathbb{E}_{\xi \sim \boldsymbol{X}_N, v} \left[ \sup_{\mathbf{w} \in \mathcal{W}} v_i \left( R_N(\mathbf{w}) - \frac{1}{m} \sum_{i=1}^{m} \ell(\mathbf{w}; \xi_i) \right) \right] \tag{31}
$$

$$
\leq 2 \mathbb{E}_v \left[ \sup_{\mathbf{w} \in \mathcal{W}} \sum_{i=1}^{m} v_i R_N(\mathbf{w}) \right] + 2 \mathbb{E}_{\xi \sim \boldsymbol{X}_N, v} \left[ \sup_{\mathbf{w} \in \mathcal{W}} \frac{1}{m} \sum_{i=1}^{m} v_i \ell(\mathbf{w}; \xi_i) \right]
$$

$$
= 2 \Re R_N(\mathbf{w}),
$$

where the first inequality holds using symmetrization inequality (see Lemma 11.4 [3]).

Recall that for any $\hat{\mathbf{w}}$, we have

$$
R_u(\hat{\mathbf{w}}) - \hat{R}_m(\hat{\mathbf{w}}) \leq \frac{N}{u} \sup_{\mathbf{w} \in \mathcal{W}} R_N(\mathbf{w}) - \hat{R}_m(\mathbf{w}).
$$

Thus, Combining (30), (31) and above inequality, for any $\delta \in (0, 1)$ with probability at least $1 - \delta$, we have

$$
R_u(\hat{\mathbf{w}}) - \hat{R}_m(\hat{\mathbf{w}}) \leq \frac{2N(1 + \eta) \Re R_N(\mathbf{w})}{u} + 4 \frac{N}{u} \sqrt{\frac{(1 + \beta) K}{m}} \log \frac{12}{\delta} + \frac{2N \sqrt{2 C^2 K^2 K_2^2 \log N \log \frac{12}{\delta}}}{mu}. \tag{32}
$$

Next, we need to bound the Rademacher complexity with traditional Dudley's integral technique. Firstly, we denote some notations. Let $d_{\mathcal{W}}(\mathbf{w}, \mathbf{w}') = \left( \frac{1}{N} \sum_{i=1}^{N} [\ell(\mathbf{w}; \mathbf{x}_i) - \ell(\mathbf{w}'; \mathbf{x}_i)]^2 \right)^{\frac{1}{2}}$. For $j \in \mathbb{N}$, let $\alpha_j = 2^{-j} M$ with $M = \sup_{\mathbf{w} \in \mathcal{W}_R} d_{\mathcal{W}}(\mathbf{w}, \mathbf{w}^{(1)})$, where $\mathcal{W}_R$ denotes the parameter space consisting of the initial parameters $\mathbf{w}^{(1)}$ together with all possible $\mathbf{w}^{(i)}$ that can be obtained using Algorithm 1. Denote by $T_j$ the minimal $\alpha_j$-cover of $\mathcal{W}_R$ and $\ell(\mathbf{w}^j; \mathbf{x})[\mathbf{w}]$ the element in $T_j$ that covers $\ell(\mathbf{w}; \mathbf{x})$. Specifically, since $\{\ell(\mathbf{w}^{(1)}; \mathbf{x})\}$ is a $M$-cover of $\mathcal{W}_R$, we set $\ell(\mathbf{w}^0; \mathbf{x})[\mathbf{w}] = \ell(\mathbf{w}^{(1)}; \mathbf{x})[\mathbf{w}]$, ( Note that $\mathbf{w}^{(1)}$ is the initialization parameter and $\mathbf{w}^j$ is the associated parameter of

$\ell$ in $T_j$). For arbitrary $n \in \mathbb{N}$:

$$
\mathbb{E}_{\boldsymbol{v}}\left[\sup_{\mathbf{w}\in\mathcal{W}_R}\sum_{i=1}^{N}v_i\ell(\mathbf{w};\mathbf{x}_i)\right]
$$

$$
=\mathbb{E}_{\boldsymbol{v}}\left[\sup_{\mathbf{w}\in\mathcal{W}_R}\left(\sum_{i=1}^{N}\Big(v_i(\ell(\mathbf{w};\mathbf{x}_i)-\ell(\mathbf{w}^n;\mathbf{x}_i))[\mathbf{w}]\right.\right.
$$

$$
\left.\left.+\sum_{j=1}^{n}v_i(\ell(\mathbf{w}^j;\mathbf{x}_i)[\mathbf{w}]-\ell(\mathbf{w}^{j-1};\mathbf{x}_i)[\mathbf{w}])+v_i\ell(\mathbf{w}^{(1)};\mathbf{x}_i)\Big)\right)\right] \tag{33}
$$

$$
\leq\mathbb{E}_{\boldsymbol{v}}\left[\sup_{\mathbf{w}\in\mathcal{W}_R}\left(\sum_{i=1}^{N}v_i(\ell(\mathbf{w};\mathbf{x}_i)-\ell(\mathbf{w}^n;\mathbf{x}_i)[\mathbf{w}])\right)\right]+\mathbb{E}_{\boldsymbol{v}}\left[\sum_{i=1}^{N}v_i\ell(\mathbf{w}^{(1)};\mathbf{x}_i)\right]
$$

$$
+\sum_{j=1}^{n}\mathbb{E}_{\boldsymbol{v}}\left[\sup_{\mathbf{w}\in\mathcal{W}_R}\left(\sum_{i=1}^{N}v_i(\ell(\mathbf{w}^j;\mathbf{x}_i)[\mathbf{w}]-\ell(\mathbf{w}^{j-1};\mathbf{x}_i)[\mathbf{w}])\right)\right].
$$

For the first term, we apply Cauchy-Schwarz inequality and obtain

$$
\mathbb{E}_{\boldsymbol{v}}\left[\sup_{\mathbf{w}\in\mathcal{W}_R}\left(\sum_{i=1}^{N}v_i(\ell(\mathbf{w};\mathbf{x}_i)-\ell(\mathbf{w}^n;\mathbf{x}_i)[\mathbf{w}])\right)\right]
$$

$$
\leq\left(\mathbb{E}_{\boldsymbol{v}}\left[\sum_{i=1}^{N}v_i^2\right]\right)^{\frac{1}{2}}\left(\sup_{\mathbf{w}\in\mathcal{W}_R}\sum_{i=1}^{N}(\ell(\mathbf{w};\mathbf{x}_i)-\ell(\mathbf{w}^n;\mathbf{x}_i)[\mathbf{w}])^2\right)^{\frac{1}{2}}\leq N\alpha_n. \tag{34}
$$

By Massart's Lemma, we have

$$
\mathbb{E}_{\boldsymbol{v}}\left[\sup_{\mathbf{w}\in\mathcal{W}_R}\left(\sum_{i=1}^{N}v_i(\ell(\mathbf{w}^j;\mathbf{x}_i)[\mathbf{w}]-\ell(\mathbf{w}^{j-1};\mathbf{x}_i)[\mathbf{w}])\right)\right]
$$

$$
\leq\sqrt{N}\sup_{\mathbf{w}\in\mathcal{W}_R}d_{\mathcal{W}}(\mathbf{w}^j,\mathbf{w}^{j-1})\sqrt{2\log|T_j||T_{j-1}|}. \tag{35}
$$

By the Minkowski inequality,

$$
\sup_{\mathbf{w}\in\mathcal{W}_R}d_{\mathcal{W}}(\mathbf{w}^j,\mathbf{w}^{j-1})
$$

$$
=\sup_{\mathbf{w}\in\mathcal{W}_R}\left(\frac{1}{N}\sum_{i=1}^{N}\left[\ell(\mathbf{w}^j;\mathbf{x}_i)[\mathbf{w}]-\ell(\mathbf{w};\mathbf{x})+\ell(\mathbf{w};\mathbf{x})-\ell(\mathbf{w}^{j-1};\mathbf{x}_i)[\mathbf{w}]\right]^2\right)^{\frac{1}{2}}
$$

$$
\leq\sup_{\mathbf{w}\in\mathcal{W}_R}\left(\frac{1}{N}\sum_{i=1}^{N}\left[\ell(\mathbf{w}^j;\mathbf{x}_i)[\mathbf{w}]-\ell(\mathbf{w};\mathbf{x})\right]^2\right)^{\frac{1}{2}} \tag{36}
$$

$$
+\sup_{\mathbf{w}\in\mathcal{W}_R}\left(\frac{1}{N}\sum_{i=1}^{N}\left[\ell(\mathbf{w};\mathbf{x})-\ell(\mathbf{w}^{j-1};\mathbf{x}_i)[\mathbf{w}]\right]^2\right)^{\frac{1}{2}}
$$

$$
=\sup_{\mathbf{w}\in\mathcal{W}_R}d_{\mathcal{W}}(\mathbf{w}^j,\mathbf{w})+\sup_{\mathbf{w}\in\mathcal{W}_R}d_{\mathcal{W}}(\mathbf{w},\mathbf{w}^{j-1})\leq\alpha_j+\alpha_{j-1}=3\alpha_j.
$$

Plugging (36) into (35), using facts that $\alpha_j = 2(\alpha_j - \alpha_{j+1})$ and $|T_j| \geq |T_{j-1}|$, taking summation over $j$,

$$
\sum_{j=1}^{n} \mathbb{E}_{\boldsymbol{v}} \left[ \sup_{\mathbf{w} \in \mathcal{W}_R} \left( \sum_{i=1}^{N} v_i (\ell(\mathbf{w}^j; \mathbf{x}_i)[\mathbf{w}] - \ell(\mathbf{w}^{j-1}; \mathbf{x}_i)[\mathbf{w}]) \right) \right]
$$

$$
\leq 6\sqrt{N} \sum_{j=1}^{n} \alpha_j \sqrt{\log |T_j|} = 12\sqrt{N} \sum_{j=1}^{n} (\alpha_j - \alpha_{j+1}) \sqrt{\log |T_j|}
$$

$$
= 12\sqrt{N} \sum_{j=1}^{n} (\alpha_j - \alpha_{j+1}) \sqrt{\log \mathcal{N}(\alpha_j, \mathcal{W}_R, d_{\mathcal{W}})}
$$

$$
\leq 12\sqrt{N} \int_{\alpha_{n+1}}^{\alpha_0} \sqrt{\log \mathcal{N}(\alpha, \mathcal{W}_R, d_{\mathcal{W}})} \, d\alpha \leq 12\sqrt{N} \int_{\alpha_{n+1}}^{\infty} \sqrt{\log \mathcal{N}(\alpha, \mathcal{W}_R, d_{\mathcal{W}})} \, d\alpha.
$$

(37)

For the last term, for any $\delta \in (0, 1)$, with probability at least $1 - \frac{\delta}{2}$ we have

$$
\mathbb{E}_{\boldsymbol{v}} \left[ \sum_{i=1}^{N} v_i \ell(\mathbf{w}^{(1)}; \mathbf{x}_i) \right] \leq \left( \sum_{i=1}^{N} \ell^2(\mathbf{w}^{(1)}; \mathbf{x}_i) \right)^{\frac{1}{2}} \leq K\sqrt{N \log \frac{2}{\delta}},
$$

(38)

where $K$ is a positive constant. The first inequality holds by Khintchine-Kahane inequality [22]. The second inequality satisfies because $\ell(\cdot; \mathbf{x})$ is sub-Gaussian, therefore, $\ell(\cdot; \mathbf{x})$ is sub-exponential. Using Lemma 4, we can derive the inequality.

Taking the limit as $n \to \infty$, plugging (34), (37) and (38) into (33) and combining with the difination of Rademacher complexity, for any $\delta \in (0, 1)$, with probability at least $1 - \frac{\delta}{2}$, we have

$$
\mathfrak{R}R_N(\mathbf{w}) = \frac{1}{N} \mathbb{E}_v \left[ \sup_{\mathbf{w} \in \mathcal{W}} \sum_{i=1}^{N} v_i \ell(\mathbf{w}; \mathbf{x}_i) \right] \leq \frac{K\sqrt{\log \frac{2}{\delta}}}{\sqrt{N}} + \frac{12}{\sqrt{N}} \int_0^{\infty} \sqrt{\log \mathcal{N}(\varepsilon, \mathcal{W}_R, d_{\mathcal{W}})} \, d\varepsilon,
$$

(39)

where $v_i$ is Rademacher random variable. One can verify that $d_{\mathcal{W}_R}(\ell(\mathbf{w}; \cdot), \ell(\mathbf{w}'; \cdot)) = \max_{z \in \mathcal{Z}} |\ell(\mathbf{w}; z) - \ell(\mathbf{w}'; z)|$ is a metric in $\mathcal{W}_R$. we have

$$
d_{\mathcal{W}} \leq \left( \frac{1}{N} \sum_{i=1}^{N} \left[ \max_{\mathbf{w}, \mathbf{w}' \in \mathcal{W}_R, \mathbf{x} \in \mathcal{Z}} \ell(\mathbf{w}; z_i) - \ell(\mathbf{w}'; \mathbf{x}_i) \right]^2 \right)^{\frac{1}{2}} \leq d_{\mathcal{W}_R}.
$$

By the definition of covering number, we have $\mathcal{N}(\varepsilon, \mathcal{W}_R, d_{\mathcal{W}}) \leq \mathcal{N}(\varepsilon, \mathcal{W}_R, d_{\mathcal{W}_R})$. Besides, applying Lemma 1 yields

$$
d_{\mathcal{W}_R} = \max_{\mathbf{x} \in \mathcal{Z}} |\ell(\mathbf{w}; z) - \ell(\mathbf{w}'; z)| \leq L_{\mathcal{F}} \|\mathbf{w} - \mathbf{w}'\|_2.
$$

By the definition of covering number, we have $\mathcal{N}(\varepsilon, \mathcal{W}_R, d_{\mathcal{W}_R}) \leq \mathcal{N}\left( \frac{\varepsilon}{L_{\mathcal{F}}}, \mathcal{B}(\mathbf{w}^{(1)}, R), d_{\mathbf{w}} \right)$, where $d_{\mathbf{w}}(\mathbf{w}, \mathbf{w}') = \|\mathbf{w} - \mathbf{w}'\|_2$ and $\mathcal{W}_R \in \mathcal{B}(\mathbf{w}^{(1)}, R)$.

According to [33], $\log \mathcal{N}\left( \varepsilon, \mathcal{B}(\mathbf{w}^{(1)}, R), d_{\mathbf{w}} \right) \leq d \log(3R/\varepsilon)$ holds. Therefore, we obtain

$$
\log \mathcal{N}(\varepsilon, \mathcal{W}_R, d_{\mathcal{W}}) \leq d \log \left( \frac{3L_{\mathcal{F}}R}{\varepsilon} \right).
$$

(40)

Furthermore,

$$
d_{\mathcal{W}}^2(\mathbf{w}, \mathbf{w}^{(1)}) = \frac{1}{N} \sum_{i=1}^{N} \left[ \ell(\mathbf{w}; \mathbf{x}_i) - \ell(\mathbf{w}^{(1)}; \mathbf{x}_i) \right]^2 \leq L_{\mathcal{F}}^2 R^2,
$$

where the last inequality is due to Lemma 1. This implies that

$$
\int_0^{\infty} \sqrt{\log \mathcal{N}(\varepsilon, \mathcal{W}_R, d_{\mathcal{W}})} d\varepsilon = \int_0^{L_{\mathcal{F}}R} \sqrt{\log \mathcal{N}(\varepsilon, \mathcal{W}_R, d_{\mathcal{W}})} \, d\varepsilon.
$$

(41)

Combining (39), (40), and (41), for any $\delta \in (0,1)$, with probability at least $1 - \frac{\delta}{2}$ yields

$$
\begin{aligned}
\mathcal{R}_N(\mathbf{w}) &\leq \frac{K\sqrt{\log \frac{2}{\delta}}}{\sqrt{N}} + 12\sqrt{\frac{d}{N}} \int_0^{L_{\mathcal{F}}R} \sqrt{\log\left(3L_{\mathcal{F}}R/\varepsilon\right)}\, \mathrm{d}\varepsilon \\
&\leq \frac{K\sqrt{\log \frac{2}{\delta}}}{\sqrt{N}} + 12\sqrt{\frac{d}{N}}\left(\sqrt{\log 3} + \frac{3}{2}\sqrt{\pi}\right) L_{\mathcal{F}}R.
\end{aligned}
\tag{42}
$$

Applying Theorem 47 in [27] to bound $R$ in (42) and plugging in (32) with probability $1 - \delta/2$, we conclude that with probability at least $1 - \delta$,

$$
R_u(\mathbf{w}^{(T+1)}) - \hat{R}_m(\mathbf{w}^{(T+1)}) = \begin{cases} \mathcal{O}\left(L_{\mathcal{F}}\frac{\sqrt{Nd}}{u}\log^{\frac{1}{2}}(T)T^{\frac{1}{2}-\alpha}\log\left(\frac{1}{\delta}\right) + \frac{N\log\left(\frac{1}{\delta}\right)}{u\sqrt{m}}\right) & \text{If } \alpha \in \left(0, \frac{1}{2}\right) \\ \mathcal{O}\left(L_{\mathcal{F}}\frac{\sqrt{Nd}}{u}\log(T)\log(\frac{1}{\delta}) + \frac{N\log\left(\frac{1}{\delta}\right)}{u\sqrt{m}}\right) & \text{If } \alpha = \frac{1}{2} \\ \mathcal{O}\left(L_{\mathcal{F}}\frac{\sqrt{Nd}}{u}\log^{\frac{1}{2}}(T)\log(\frac{1}{\delta}) + \frac{N\log\left(\frac{1}{\delta}\right)}{u\sqrt{m}}\right) & \text{If } \alpha \in \left(\frac{1}{2}, 1\right]. \end{cases}
$$

The proof is complete. $\qquad\square$

*Proof of Theorem 11.* In order to obtain high-probability bounds with out new concentration inequalities, for the term $\sup_{f_{\mathbf{w}} \in \mathcal{F}_{\mathcal{W}}} \sum_{\mathbf{x} \in \mathbf{X}_m} f_{\mathbf{w}}(\mathbf{x}) = \sup_{\mathbf{w} \in \mathcal{W}} \sum_{\mathbf{x} \in \mathbf{X}_m} (R_N(\mathbf{w}) - \ell(\mathbf{w}; \mathbf{x})) = m \cdot \sup_{\mathbf{w} \in \mathcal{W}} (R_N(\mathbf{w}) - \hat{R}_m(\mathbf{w}))$, where we obtain a factor of $m$ in the equation because in Theorem 2 we considered unnormalized sums.

Then, to use Theorem 2, we need to bound $\left\|\max_{\mathbf{x}} \sup_{f_{\mathbf{w}} \in \mathcal{F}_{\mathcal{W}}} f_{\mathbf{w}}(\mathbf{x})\right\|_{\psi_1}^2$.

$$
\left\|\max_{\mathbf{x}} \sup_{f_{\mathbf{w}} \in \mathcal{F}_{\mathcal{W}}} f_{\mathbf{w}}(\mathbf{x})\right\|_{\psi_1}^2 \leq \left\|\max_{\mathbf{x}} \sup_{\mathbf{w} \in \mathcal{W}} \ell(\mathbf{w}; \mathbf{x})\right\|_{\psi_1}^2 \leq K^2 \max_{\mathbf{x}} \left\|\sup_{\mathbf{w} \in \mathcal{W}} \ell(\mathbf{w}; \mathbf{x})\right\|_{\psi_1}^2 \log^2 N \leq K^2 K_1^2 \log^2 N.
$$

where $K$ and $K_1$ are two constants. The second inequality holds using Theorem 7 [34] and the last inequality satisfies because $\ell(\cdot; \mathbf{x})$ is sub-exponential, using property of the tail bound for subexponential distribution

Then we turn to bound $\sigma_{\mathcal{W}}^2$. For any fixed $\mathbf{w} \in \mathcal{W}$ and any $\delta \in (0,1)$, with at least probability $1 - \frac{\delta}{2}$, we have

$$
\frac{1}{N} \sum_{\mathbf{x} \in \mathbf{Z}_N} (\ell(\mathbf{w}; \mathbf{x}) - R_N(\mathbf{w}))^2 = \frac{1}{N} \sum_{\mathbf{x} \in \mathbf{Z}_N} \ell(\mathbf{w}; \mathbf{x})^2 - R_N(\mathbf{w})^2 \leq \frac{1}{N} \sum_{\mathbf{x} \in \mathbf{Z}_N} \ell(\mathbf{w}; \mathbf{x})^2 \leq K \log^2 \frac{2}{\delta},
$$

where $K$ is a positive constant. the last inequality holds because $\ell(\cdot; \mathbf{x})$ is sub-exponential. Thus, $\ell^2(\cdot; \mathbf{x})$ is sub-Weibull random variable with tail parameter 2, using Lemma 4 we can derive the last inequality. Thus for any $\delta \in (0,1)$, with at least probability $1 - \frac{\delta}{2}$, we have

$$
\sigma_{\mathcal{W}}^2 = \sup_{\mathbf{w} \in \mathcal{W}} \left(\frac{1}{N} \sum_{\mathbf{x} \in \mathbf{Z}_N} (\ell(\mathbf{w}; \mathbf{x}) - R_N(\mathbf{w}))^2\right) \leq K \log^2 \frac{2}{\delta}
\tag{43}
$$

According to Theorem 2, Let $Q_m = m \cdot (R_N(\mathbf{w}) - \hat{R}_m(\mathbf{w}))$ and combined with (43). For any $\delta \in (0,1)$ with probability at least $1 - \delta$, we have

$$
\begin{aligned}
&\sup_{\mathbf{w} \in \mathcal{W}} (R_N(\mathbf{w}) - \hat{R}_m(\mathbf{w})) \\
&\leq (1+\eta)E_m + 4\sqrt{\left(\frac{(1+\beta)K\log^2 \frac{2}{\delta}}{m} + \frac{3C^2K^2K_1^2\log^2 N}{m^2}\right)\log \frac{12}{\delta}} \\
&\leq (1+\eta)E_m + 4\sqrt{\frac{(1+\beta)K\log^2 \frac{2}{\delta}\log \frac{12}{\delta}}{m}} + \frac{4\sqrt{3C^2K^2K_1^2\log^2 N \log \frac{12}{\delta}}}{m} \\
&\leq (1+\eta)E_m + 4\sqrt{\frac{(1+\beta)K\log^3 \frac{12}{\delta}}{m}} + \frac{4\sqrt{3C^2K^2K_1^2\log^2 N \log \frac{12}{\delta}}}{m}.
\end{aligned}
\tag{44}
$$

where the second inequality holds using $\sqrt{a+b} \leq \sqrt{a} + \sqrt{b}$.

Next, we need to bound the $E_m = \mathbb{E}\left[\sup_{\mathbf{w}\in\mathcal{W}} \left(R_N(\mathbf{w}) - \frac{1}{m}\sum_{i=1}^m \ell(\mathbf{w};\xi_i)\right)\right]$. We have

$$
\begin{aligned}
E_m &= \mathbb{E}\left[\sup_{\mathbf{w}\in\mathcal{W}} \left(R_N(\mathbf{w}) - \frac{1}{m}\sum_{i=1}^m \ell(\mathbf{w};\xi_i)\right)\right] \\
&\leq 2\mathbb{E}_{\xi\sim\boldsymbol{X}_N,v}\left[\sup_{\mathbf{w}\in\mathcal{W}} v_i\left(R_N(\mathbf{w}) - \frac{1}{m}\sum_{i=1}^m \ell(\mathbf{w};\xi_i)\right)\right] \\
&\leq 2\mathbb{E}_v\left[\sup_{\mathbf{w}\in\mathcal{W}} \sum_{i=1}^m v_i R_N(\mathbf{w})\right] + 2\mathbb{E}_{\xi\sim\boldsymbol{X}_N,v}\left[\sup_{\mathbf{w}\in\mathcal{W}} \frac{1}{m}\sum_{i=1}^m v_i\ell(\mathbf{w};\xi_i)\right] \\
&= 2\Re R_N(\mathbf{w}),
\end{aligned}
\tag{45}
$$

where the first inequality holds using symmetrization inequality (see Lemma 11.4 [3]).

Recall that for any $\hat{\mathbf{w}}$, we have

$$
R_u(\hat{\mathbf{w}}) - \hat{R}_m(\hat{\mathbf{w}}) \leq \frac{N}{u}\sup_{\mathbf{w}\in\mathcal{W}} R_N(\mathbf{w}) - \hat{R}_m(\mathbf{w}).
$$

Thus, Combining (44), (45) and above inequality, for any $\delta \in (0,1)$ with probability at least $1-\delta$, we have

$$
R_u(\hat{\mathbf{w}}) - \hat{R}_m(\hat{\mathbf{w}}) \leq \frac{2N(1+\eta)\Re R_N}{u} + 4\frac{N}{u}\sqrt{\frac{(1+\beta)K\log^3\frac{12}{\delta}}{m}} + \frac{4N\sqrt{3C^2K^2K_1^2\log^2 N \log\frac{12}{\delta}}}{mu}.
\tag{46}
$$

Next, we need to bound the Rademacher complexity with traditional Dudley's integral technique. Let $d_{\mathcal{W}}(\mathbf{w},\mathbf{w}') = \left(\frac{1}{N}\sum_{i=1}^N [\ell(\mathbf{w};\mathbf{x}_i) - \ell(\mathbf{w}';\mathbf{x}_i)]^2\right)^{\frac{1}{2}}$. For $j\in\mathbb{N}$, let $\alpha_j = 2^{-j}M$ with $M = \sup_{\mathbf{w}\in\mathcal{W}_R} d_{\mathcal{W}}(\mathbf{w},\mathbf{w}^{(1)})$, where $\mathcal{W}_R$ denotes the parameter space consisting of the initial parameters $\mathbf{w}^{(1)}$ together with all possible $\mathbf{w}^{(i)}$ that can be obtained using Algorithm 1. Denote by $T_j$ the minimal $\alpha_j$-cover of $\mathcal{W}_R$ and $\ell(\mathbf{w}^j;\mathbf{x})[\mathbf{w}]$ the element in $T_j$ that covers $\ell(\mathbf{w};\mathbf{x})$. Specifically, since $\{\ell(\mathbf{w}^{(1)};\mathbf{x})\}$ is a $M$-cover of $\mathcal{W}_R$, we set $\ell(\mathbf{w}^0;\mathbf{x})[\mathbf{w}] = \ell(\mathbf{w}^{(1)};\mathbf{x})[\mathbf{w}]$, ( Note that $\mathbf{w}^{(1)}$ is the initialization parameter and $\mathbf{w}^j$ is the associated parameter of $\ell$ in $T_j$). For arbitrary $n\in\mathbb{N}$:

$$
\begin{aligned}
&\mathbb{E}_{\boldsymbol{v}}\left[\sup_{\mathbf{w}\in\mathcal{W}_R} \sum_{i=1}^N v_i\ell(\mathbf{w};\mathbf{x}_i)\right] \\
&= \mathbb{E}_{\boldsymbol{v}}\left[\sup_{\mathbf{w}\in\mathcal{W}_R} \left(\sum_{i=1}^N \left(v_i(\ell(\mathbf{w};\mathbf{x}_i) - \ell(\mathbf{w}^n;\mathbf{x}_i))[\mathbf{w}]\right.\right.\right. \\
&\qquad\qquad \left.\left.\left. + \sum_{j=1}^n v_i(\ell(\mathbf{w}^j;\mathbf{x}_i)[\mathbf{w}] - \ell(\mathbf{w}^{j-1};\mathbf{x}_i)[\mathbf{w}]) + v_i\ell(\mathbf{w}^{(1)};\mathbf{x}_i)\right)\right)\right] \\
&\leq \mathbb{E}_{\boldsymbol{v}}\left[\sup_{\mathbf{w}\in\mathcal{W}_R} \left(\sum_{i=1}^N v_i(\ell(\mathbf{w};\mathbf{x}_i) - \ell(\mathbf{w}^n;\mathbf{x}_i)[\mathbf{w}])\right)\right] + \mathbb{E}_{\boldsymbol{v}}\left[\sum_{i=1}^N v_i\ell(\mathbf{w}^{(1)};\mathbf{x}_i)\right] \\
&\qquad + \sum_{j=1}^n \mathbb{E}_{\boldsymbol{v}}\left[\sup_{\mathbf{w}\in\mathcal{W}_R} \left(\sum_{i=1}^N v_i(\ell(\mathbf{w}^j;\mathbf{x}_i)[\mathbf{w}] - \ell(\mathbf{w}^{j-1};\mathbf{x}_i)[\mathbf{w}])\right)\right].
\end{aligned}
\tag{47}
$$

For the first term, we apply Cauchy-Schwarz inequality and obtain

$$
\begin{aligned}
&\mathbb{E}_{\boldsymbol{v}}\left[\sup_{\mathbf{w}\in\mathcal{W}_R} \left(\sum_{i=1}^N v_i(\ell(\mathbf{w};\mathbf{x}_i) - \ell(\mathbf{w}^n;\mathbf{x}_i)[\mathbf{w}])\right)\right] \\
&\leq \left(\mathbb{E}_{\boldsymbol{v}}\left[\sum_{i=1}^N v_i^2\right]\right)^{\frac{1}{2}} \left(\sup_{\mathbf{w}\in\mathcal{W}_R} \sum_{i=1}^N (\ell(\mathbf{w};\mathbf{x}_i) - \ell(\mathbf{w}^n;\mathbf{x}_i)[\mathbf{w}])^2\right)^{\frac{1}{2}} \leq N\alpha_n.
\end{aligned}
\tag{48}
$$

By Massart's Lemma, we have

$$
\mathbb{E}_{\boldsymbol{v}}\left[\sup_{\mathbf{w}\in\mathcal{W}_R}\left(\sum_{i=1}^{N}v_i(\ell(\mathbf{w}^j;\mathbf{x}_i)[\mathbf{w}]-\ell(\mathbf{w}^{j-1};\mathbf{x}_i)[\mathbf{w}])\right)\right]
$$
$$
\leq\sqrt{N}\sup_{\mathbf{w}\in\mathcal{W}_R}d_{\mathcal{W}}(\mathbf{w}^j,\mathbf{w}^{j-1})\sqrt{2\log|T_j||T_{j-1}|}.
$$
(49)

By the Minkowski inequality,

$$
\sup_{\mathbf{w}\in\mathcal{W}_R}d_{\mathcal{W}}(\mathbf{w}^j,\mathbf{w}^{j-1})
$$
$$
=\sup_{\mathbf{w}\in\mathcal{W}_R}\left(\frac{1}{N}\sum_{i=1}^{N}\left[\ell(\mathbf{w}^j;\mathbf{x}_i)[\mathbf{w}]-\ell(\mathbf{w};\mathbf{x})+\ell(\mathbf{w};\mathbf{x})-\ell(\mathbf{w}^{j-1};\mathbf{x}_i)[\mathbf{w}]\right]^2\right)^{\frac{1}{2}}
$$
$$
\leq\sup_{\mathbf{w}\in\mathcal{W}_R}\left(\frac{1}{N}\sum_{i=1}^{N}\left[\ell(\mathbf{w}^j;\mathbf{x}_i)[\mathbf{w}]-\ell(\mathbf{w};\mathbf{x})\right]^2\right)^{\frac{1}{2}}+\sup_{\mathbf{w}\in\mathcal{W}_R}\left(\frac{1}{N}\sum_{i=1}^{N}\left[\ell(\mathbf{w};\mathbf{x})-\ell(\mathbf{w}^{j-1};\mathbf{x}_i)[\mathbf{w}]\right]^2\right)^{\frac{1}{2}}
$$
$$
=\sup_{\mathbf{w}\in\mathcal{W}_R}d_{\mathcal{W}}(\mathbf{w}^j,\mathbf{w})+\sup_{\mathbf{w}\in\mathcal{W}_R}d_{\mathcal{W}}(\mathbf{w},\mathbf{w}^{j-1})\leq\alpha_j+\alpha_{j-1}=3\alpha_j.
$$
(50)

Plugging (50) into (49), using facts that $\alpha_j=2(\alpha_j-\alpha_{j+1})$ and $|T_j|\geq|T_{j-1}|$, taking summation over $j$,

$$
\sum_{j=1}^{n}\mathbb{E}_{\boldsymbol{v}}\left[\sup_{\mathbf{w}\in\mathcal{W}_R}\left(\sum_{i=1}^{N}v_i(\ell(\mathbf{w}^j;\mathbf{x}_i)[\mathbf{w}]-\ell(\mathbf{w}^{j-1};\mathbf{x}_i)[\mathbf{w}])\right)\right]
$$
$$
\leq6\sqrt{N}\sum_{j=1}^{n}\alpha_j\sqrt{\log|T_j|}=12\sqrt{N}\sum_{j=1}^{n}(\alpha_j-\alpha_{j+1})\sqrt{\log|T_j|}
$$
$$
=12\sqrt{N}\sum_{j=1}^{n}(\alpha_j-\alpha_{j+1})\sqrt{\log\mathcal{N}(\alpha_j,\mathcal{W}_R,d_{\mathcal{W}})}
$$
(51)
$$
\leq12\sqrt{N}\int_{\alpha_{n+1}}^{\alpha_0}\sqrt{\log\mathcal{N}(\alpha,\mathcal{W}_R,d_{\mathcal{W}})}\,\mathrm{d}\alpha
$$
$$
\leq12\sqrt{N}\int_{\alpha_{n+1}}^{\infty}\sqrt{\log\mathcal{N}(\alpha,\mathcal{W}_R,d_{\mathcal{W}})}\,\mathrm{d}\alpha.
$$

For the last term, for any $\delta\in(0,1)$, with probability at least $1-\frac{\delta}{2}$ we have

$$
\mathbb{E}_{\boldsymbol{v}}\left[\sum_{i=1}^{N}v_i\ell(\mathbf{w}^{(1)};\mathbf{x}_i)\right]\leq\left(\sum_{i=1}^{N}\ell^2(\mathbf{w}^{(1)};\mathbf{x}_i)\right)^{\frac{1}{2}}\leq K\sqrt{N}\log\frac{2}{\delta},
$$
(52)

where $K$ is a positive constant. The first inequality holds by Khintchine-Kahane inequality [22]. The second inequality satisfies because $\ell(\cdot;\mathbf{x})$ is sub-exponential, therefore, $\ell^2(\cdot;\mathbf{x})$ is sub-weibull random variables with parameter 2. Using Lemma 4, we can derive the inequality.

Taking the limit as $n\to\infty$, plugging (48), (51) and (52) into (47) and combining with the difination of Rademacher complexity, for any $\delta\in(0,1)$, with probability at least $1-\frac{\delta}{2}$, we have

$$
\mathfrak{R}R_N(\mathbf{w})=\frac{1}{N}\mathbb{E}_v\left[\sup_{\mathbf{w}\in\mathcal{W}}\sum_{i=1}^{N}v_i\ell(\mathbf{w};\mathbf{x}_i)\right]\leq\frac{K\log\frac{2}{\delta}}{\sqrt{N}}+\frac{12}{\sqrt{N}}\int_0^{\infty}\sqrt{\log\mathcal{N}(\varepsilon,\mathcal{W}_R,d_{\mathcal{W}})}\,\mathrm{d}\varepsilon,
$$
(53)

where $v_i$ is Rademacher random variable. One can verify that $d_{\mathcal{W}_R}(\ell(\mathbf{w};\cdot),\ell(\mathbf{w}';\cdot))=\max_{z\in\mathcal{Z}}|\ell(\mathbf{w};z)-\ell(\mathbf{w}';z)|$ is a metric in $\mathcal{W}_R$. we have

$$
d_{\mathcal{W}}\leq\left(\frac{1}{N}\sum_{i=1}^{N}\left[\max_{\mathbf{w},\mathbf{w}'\in\mathcal{W}_R,\mathbf{x}\in\mathcal{Z}}\ell(\mathbf{w};z_i)-\ell(\mathbf{w}';\mathbf{x}_i)\right]^2\right)^{\frac{1}{2}}\leq d_{\mathcal{W}_R}.
$$

By the definition of covering number, we have $\mathcal{N}(\varepsilon, \mathcal{W}_R, d_{\mathcal{W}}) \leq \mathcal{N}(\varepsilon, \mathcal{W}_R, d_{\mathcal{W}_R})$. Besides, applying Lemma 1 yields

$$d_{\mathcal{W}_R} = \max_{\mathbf{x} \in \mathcal{Z}} |\ell(\mathbf{w}; z) - \ell(\mathbf{w}'; z)| \leq L_{\mathcal{F}} \|\mathbf{w} - \mathbf{w}'\|_2.$$

By the definition of covering number, we have $\mathcal{N}(\varepsilon, \mathcal{W}_R, d_{\mathcal{W}_R}) \leq \mathcal{N}\left(\frac{\varepsilon}{L_{\mathcal{F}}}, \mathcal{B}(\mathbf{w}^{(1)}, R), d_{\mathbf{w}}\right)$, where $d_{\mathbf{w}}(\mathbf{w}, \mathbf{w}') = \|\mathbf{w} - \mathbf{w}'\|_2$ and $\mathcal{W}_R \in \mathcal{B}(\mathbf{w}^{(1)}, R)$.

According to [33], $\log \mathcal{N}\left(\varepsilon, \mathcal{B}(\mathbf{w}^{(1)}, R), d_{\mathbf{w}}\right) \leq d \log(3R/\varepsilon)$ holds. Therefore, we obtain

$$\log \mathcal{N}(\varepsilon, \mathcal{W}_R, d_{\mathcal{W}}) \leq d \log\left(\frac{3L_{\mathcal{F}} R}{\varepsilon}\right). \tag{54}$$

Furthermore,

$$d_{\mathcal{W}}^2(\mathbf{w}, \mathbf{w}^{(1)}) = \frac{1}{N} \sum_{i=1}^{N} \left[\ell(\mathbf{w}; \mathbf{x}_i) - \ell(\mathbf{w}^{(1)}; \mathbf{x}_i)\right]^2 \leq L_{\mathcal{F}}^2 R^2,$$

where the last inequality is due to Lemma 1. This implies that

$$\int_0^\infty \sqrt{\log \mathcal{N}(\varepsilon, \mathcal{W}_R, d_{\mathcal{W}})} \mathrm{d}\varepsilon = \int_0^{L_{\mathcal{F}} R} \sqrt{\log \mathcal{N}(\varepsilon, \mathcal{W}_R, d_{\mathcal{W}})} \, \mathrm{d}\varepsilon. \tag{55}$$

Combining (53), (54), and (55), for any $\delta \in (0, 1)$, with probability at least $1 - \frac{\delta}{2}$ yields

$$\begin{aligned}
\mathcal{R}_N(\mathbf{w}) &\leq \frac{K \log \frac{2}{\delta}}{\sqrt{N}} + 12 \sqrt{\frac{d}{N}} \int_0^{L_{\mathcal{F}} R} \sqrt{\log(3L_{\mathcal{F}} R/\varepsilon)} \, \mathrm{d}\varepsilon \\
&\leq \frac{K \log \frac{2}{\delta}}{\sqrt{N}} + 12 \sqrt{\frac{d}{N}} \left(\sqrt{\log 3} + \frac{3}{2}\sqrt{\pi}\right) L_{\mathcal{F}} R.
\end{aligned} \tag{56}$$

Applying Theorem 47 in [27] to bound $R$ in (56) and plugging in (46) with probability $1 - \delta/2$, we conclude that with probability at least $1 - \delta$,

$$R_u(\mathbf{w}^{(T+1)}) - \hat{R}_m(\mathbf{w}^{(T+1)}) = \begin{cases} \mathcal{O}\left(L_{\mathcal{F}} \frac{\sqrt{Nd}}{u} \log^{\frac{1}{2}}(T) T^{\frac{1}{2}-\alpha} \log\left(\frac{1}{\delta}\right) + \frac{N}{u} \sqrt{\frac{\log^3\left(\frac{1}{\delta}\right)}{m}}\right) & \text{If } \alpha \in \left(0, \frac{1}{2}\right) \\ \mathcal{O}\left(L_{\mathcal{F}} \frac{\sqrt{Nd}}{u} \log(T) \log(\frac{1}{\delta}) + \frac{N}{u} \sqrt{\frac{\log^3\left(\frac{1}{\delta}\right)}{m}}\right) & \text{If } \alpha = \frac{1}{2} \\ \mathcal{O}\left(L_{\mathcal{F}} \frac{\sqrt{Nd}}{u} \log^{\frac{1}{2}}(T) \log(\frac{1}{\delta}) + \frac{N}{u} \sqrt{\frac{\log^3\left(\frac{1}{\delta}\right)}{m}}\right) & \text{If } \alpha \in \left(\frac{1}{2}, 1\right]. \end{cases}$$

The proof is complete. $\qquad\qquad\square$

There is nothing special about the proofs of Corollary 1 and Corollary 2, which simply involve combining Theorem 5 (or Theorem 11) with an existing optimization result. Here we give the proof of Corollary 1 as an example.

*Proof of Corollary 1.* By Lemma 43 in [27], we have

$$\hat{R}_m(\mathbf{w}^{T+1}) - \hat{R}_m(\hat{\mathbf{w}}^*) = \begin{cases} \mathcal{O}\left(\frac{1}{T^\alpha}\right) & \text{if } \alpha \in (0, 1) \\ \mathcal{O}\left(\frac{\log(T) \log^3(1/\delta)}{T}\right) & \text{if } \alpha = 1. \end{cases} \tag{57}$$

By Theorem 5,

$$R_u(\mathbf{w}^{(T+1)}) - \hat{R}_m(\mathbf{w}^{(T+1)}) = \begin{cases} \mathcal{O}\left(L_{\mathcal{F}} \frac{\sqrt{Nd}}{u} \log^{\frac{1}{2}}(T) T^{\frac{1}{2}-\alpha} \log\left(\frac{1}{\delta}\right) + \frac{N \log\left(\frac{1}{\delta}\right)}{u\sqrt{m}}\right) & \text{If } \alpha \in \left(0, \frac{1}{2}\right) \\ \mathcal{O}\left(L_{\mathcal{F}} \frac{\sqrt{Nd}}{u} \log(T) \log(\frac{1}{\delta}) + \frac{N \log\left(\frac{1}{\delta}\right)}{u\sqrt{m}}\right) & \text{If } \alpha = \frac{1}{2} \\ \mathcal{O}\left(L_{\mathcal{F}} \frac{\sqrt{Nd}}{u} \log^{\frac{1}{2}}(T) \log(\frac{1}{\delta}) + \frac{N \log\left(\frac{1}{\delta}\right)}{u\sqrt{m}}\right) & \text{If } \alpha \in \left(\frac{1}{2}, 1\right]. \end{cases} \tag{58}$$

Combing (57) and (58) yields the result. $\qquad\qquad\square$

