# OpenReview forum: "Improved Risk Bounds with Unbounded Losses for Transductive Learning"
_ICLR.cc/2025/Conference — Submitted to ICLR 2025_

### Official Review · Reviewer_8kPJ · 2024-10-26

**Soundness:** 1
**Presentation:** 2
**Contribution:** 1
**Rating:** 1
**Confidence:** 5

**Summary:**

The paper studies potentially improved risk bound for transductive learning following the conventional localized method. The main difference the author claims is that the risk bounds are for unbounded functions. However, such claim, together with the technical results for  unbounded functions, are very questionable. Furthermore, there are no detailed comparison to the current state-of-the-art risk bounds for the main results in Theorem 3 and Theorem 4.

**Strengths:**

The paper studies potentially improved risk bound for transductive learning following the conventional localized method. The main difference the author claims is that the risk bounds are for unbounded functions. However, such claim, together with the technical results for  unbounded functions, are very questionable.

**Weaknesses:**

There are several major technical drawbacks.



1. While this paper claims that the risk bounds for transductive learning are for unbounded loss functions, the assumptions required for the results are essentially designed for bounded loss functions. For example, the main results Theorem 3 and Theorem 4 need the assumption that $E[f^2] \le B E[f]$. It is well known that such assumption, $E[f^2] \le B E[f]$, holds mainly for bounded loss functions, such as that in the classical local Rademacher complexity work (Bartlett Local Rademacher Complexities, AOS 2005). It turns out that while the paper claims risk bounds for "unbounded loss", but the results rely on the assumption which mainly hold for bounded loss functions.

2. It is well known that Rademacher complexity or local Rademacher complexity based methods derive distribution-free risk bounds that do not need distributional assumptions. In contrast, the risk bounds in the main results Theorem 3 and Theorem 4 require sub-Gaussian and sub-exponential loss functions. It is not clear which loss functions are  sub-Gaussian or sub-exponential, and such restriction on the loss functions can significantly limit the application scope of the derived bounds.

3. There are no detailed comparison to the current state-of-the-art risk bounds for the main results in Theorem 3 and Theorem 4, such as the existing transductive bounds in (Tolstikhin et al. 2014, Localized Complexities for Transductive Learning. COLT 2014). Without comparison to prior art, the significance of these results is not clear and questionable.

4. The risk bounds in the main results, Theorem 3 and Theorem 4, do not convergence to 0 under the case that $m = N^{\alpha}$ or
$m = N^{\alpha}$ with $\alpha \in (0,1/2]$, and they even diverge to $\infty$ if $\alpha \in (0,1/2)$.  This is in a strong contrast to existing risk bounds for excess risk bounds where such bounds should always at least converge to $0$, and it is really misleading to claim such risk bounds are improved ones.

**Questions:**

See weaknesses.

---

### Official Review · Reviewer_DEbX · 2024-10-31

**Soundness:** 2
**Presentation:** 2
**Contribution:** 2
**Rating:** 3
**Confidence:** 3

**Summary:**

In this work, the authors establish generalization bounds for the transductive setting, applicable even in cases with unbounded loss. The core technical contribution is a novel tail bound for the relevant empirical process. Using these results, the authors then derive generalization bounds for graph neural networks.

**Strengths:**

The transductive setting has gained renewed attention recently, as many practical problems are better suited to transductive learning than the traditional iid statistical framework. In this context, the work is particularly relevant.

The concentration bounds presented are non-trivial, and their proof involves sophisticated mathematical tools.

**Weaknesses:**

1. The authors do not offer any motivation for addressing unbounded loss. A discussion on why this is an important and relevant problem to study would be beneficial.

2. It is unclear what the significance of Theorems 3 and 4 is. Let’s consider Theorem 3 as an example. Given other terms, the upper bound can at best be
$$ \frac{N^2 \log\left( \frac{1}{\delta}\right)}{m^2 u}.$$
The authors claim this bound is state-of-the-art for $m = o(N^{2/5})$. If we set $m = N^{1/5} = o(N^{2/5})$, then $u = N - N^{1/5} \leq N $, and the upper bound is at least
$$ \geq \frac{N^2 \log\left( \frac{1}{\delta}\right)}{N^{2/5} \cdot N} \geq N^{3/5} \log\left( \frac{1}{\delta}\right).$$

Given that this is the highest the proven upperbound can be, it is difficult to see why such a bound would be of interest as $N$ can be quite large, making the bound potentially vacuous. I may likely be missing something here, and I would be happy to engage with the authors during the discussion session to gain further clarity and adjust my score.

**Questions:**

1. Given the claim that the authors prove new results for unbounded loss, it would be helpful to discuss what is already known in the context of bounded loss. Are similar concentration inequalities established for the bounded loss setting? The paper references [1,12] as proving some tail bounds—could the authors provide a brief summary of the results from those works?



2. See the weakness mentioned above. Could the authors offer a clearer explanation of how to interpret Theorems 3 and 4? Specifically, how should the generalization bound behave as the size of the training set $m$ increases? Does the bound vanish in any meaningful way as $m$ increases, and if so, how?

---

### Official Review · Reviewer_Fmcx · 2024-10-31

**Soundness:** 3
**Presentation:** 2
**Contribution:** 3
**Rating:** 8
**Confidence:** 3

**Summary:**

The paper studies the transductive learning problem, where the learner receives a subset of labeled samples drawn without replacement from a dataset, alongside unlabeled samples for which the goal is to predict the labels. As the samples are not independent and the loss function may be unbounded, the authors develop concentration inequalities for the supremum of empirical processes sampled without replacement for unbounded functions. They use these inequalities to derive tighter risk bounds for transductive learning problems and graph neural networks.

**Strengths:**

This paper is the first to derive concentration inequalities for the supremum of empirical processes sampled without replacement for unbounded functions, presenting a novel result. Furthermore, these concentration inequalities are utilized to refine the risk bounds for transductive learning and graph neural networks found in the literature.

**Weaknesses:**

The paper lacks numerical results to support the derived risk bounds. Additionally, as it does not provide lower bounds for the risk, it remains unclear whether the resulting bounds could be further improved.

There are some typos in the paper:

Line 221: we mainly "follows"
Line 222: we "introduced"
Line 405: w_1^{T+1}   ->   w^{T+1}

**Questions:**

In Theorem 1, for a fixed $c$ from the set $C$, both $f(c)$ and supremum of $f(c)$ are not random. So how is the Orlicz defined when there is no randomness here?

---

### Official Review · Reviewer_v5Gq · 2024-11-02

**Soundness:** 2
**Presentation:** 2
**Contribution:** 1
**Rating:** 1
**Confidence:** 5

**Summary:**

This paper derives risk bounds for transductive learning scenarios with specific applications to Graph Neural Networks (GNNs) under unbounded loss functions. The work focuses on theoretical guarantees for both sub-Gaussian and sub-exponential loss functions.

**Strengths:**

- Novel analysis of unbounded loss functions in the transductive learning setting
- Mathematical rigour in deriving the theoretical bounds
- Practical applications to GNN scenarios

**Weaknesses:**

**Weaknesses:**

- Limited scope of unbounded loss functions:

  - Analysis is restricted to sub-Gaussian and sub-exponential functions (and sub-Weibull in appendix)
  - Other important classes of unbounded loss functions are not addressed


- Insufficient comparison with prior work:

  - The paper overlooks crucial related work, particularly [1] (Maurer & Pontil, 2021). While [1] focuses on inductive settings, their theoretical foundations appear relevant. A comparative analysis between Theorems 1 and 2 and the results in [1] is needed.

- Limited Contribution:
  - The current contribution of theoretical analysis in the GNN framework is limited. The current results are general and independent of the Graph properties.

**Minor Comments:**

-  In Assumption 3, "α-Hölder" is misspelled
- Add the explanation of Hoeffding's reduction method to the appendix
- Use "Boundedness" instead of "Boundness"
- Use "techniques" instead of "technologies"
- Line 221 "We mainly follows the traditional technique..." --> "We mainly follow the traditional technique"

---

**References:**

- [1] Maurer, A., & Pontil, M. (2021). Concentration inequalities under sub-gaussian and sub-exponential conditions. Advances in Neural Information Processing Systems, 34, 7588-7597.

**Questions:**

- Graph Properties:

  - Which specific graph properties are leveraged in your analysis?
  - The current bounds seem applicable to general transductive learning scenarios - how are they specialized for graph-based problems?


- Asymptotic Behavior:

  - Please elaborate on the behavior of your bounds as $u \rightarrow \infty$ and $m\ll u$.

---

### Meta-Review · Area_Chair_RVBg · 2024-12-20

**Metareview:**

The paper studies the problem of obtaining risk bounds for transductive learning for bounded losses. The problem has interesting mathematical aspects, but there were several concerns pointed out by the reviewers. In particular, the paper does not provide adequate comparison to some related work, and the significance of the bounds is also not fully clear (it still requires a variance condition on the loss, and the regime in which it provides improved bounds is not clear). Therefore, the paper is not ready for acceptance.

**Additional Comments On Reviewer Discussion:**

There was no rebuttal from the authors.

---

### Decision · Program_Chairs · 2025-01-22

Reject